# Model-based Climatology of Diurnal Variability in Stratospheric Ozone as a Data Analysis Tool

Stacey M. Frith[1], Pawan K. Bhartia[2], Luke D. Oman[2], Natalya A. Kramarova[2], Richard D. McPeters[2], Gordon J. Labow[1]

[1]Science Systems and Applications, Inc., Lanham, MD, USA
[2]NASA Goddard Space Flight Center, Greenbelt, MD, USA

*Correspondence to*: Stacey M. Frith (Stacey.Frith@nasa.gov)

**Abstract.** Observational studies of stratospheric ozone often involve data from multiple instruments that measure the ozone at different times of day. There has been an increased awareness of the potential impact of the diurnal cycle when interpreting measurements of stratospheric ozone at altitudes in the mid to upper stratosphere. To address this issue, we present a climatological representation of diurnal variations in ozone with a half hour temporal resolution as a function of latitude, pressure and month, based on output from the Goddard Earth Observing System (GEOS) general circulation model coupled to the NASA Global Modeling Initiative (GMI) chemistry package (known as the GEOS-GMI chemistry model). This climatology can be applied to a wide range of ozone data analyses, including data inter-comparisons, data merging, and analysis of data from a single platform in a non-sun-synchronous orbit. We evaluate the diurnal climatology by comparing mean differences between ozone measurements made at different local solar times to the differences predicted by the diurnal model. The ozone diurnal cycle is a complicated function of latitude, pressure and season, with variations of less than 5% in the tropics and sub-tropics, increasing to more than 15% near the polar day terminator in the upper stratosphere. These results compare well with previous modeling simulations and are supported by similar size variations in satellite observations. We present several example applications of the climatology in currently relevant data studies. We also compare this diurnal climatology to the diurnal signal from a previous iteration of the free-running GEOS Chemistry Climate Model (GEOSCCM) and to the ensemble runs of GEOS-GMI to test the sensitivity of the model diurnal cycle to changes in model formulation and simulated time period.

# 1 Introduction

Stratospheric ozone has been an environmental concern since the suggestion 45 years ago that anthropogenic chemicals (collectively known as ozone depleting substances; ODSs) released into the atmosphere could destroy ozone [Stolarski and Cicerone, 1974; Molina and Rowland, 1974]. Since that time, our understanding of ozone chemistry and dynamics has vastly evolved, and key to that evolution have been high quality satellite and ground-based observations of ozone. These observations were used to quantify ozone loss during the 1980s and early 1990s, and now are being used to quantify the turn around and expected increase in ozone after the ban of many ODSs. However, the slow decline in these chemicals, resulting from their long atmospheric lifetimes and the staged reduction of their use through the Montreal Protocol and subsequent amendments, means that the ozone recovery rate will be much slower than the loss rate. Therefore, observations must be sufficiently stable to resolve these small changes in time. Furthermore, measurements from more than one source are required to provide adequate spatial and time coverage to evaluate the full range of effects of ODSs on ozone, such that data must be consistent across multiple observation platforms.

Inter-comparison of ozone observations from satellite and ground-based data sources is key to validating independent measurements and maintaining high quality data records. With the need for more stable long-term records, we must consider ever-smaller sources of variability. One such variation is the diurnal cycle in ozone, which had generally been considered small enough to be inconsequential in the middle stratosphere, though the large variations in the upper stratosphere and mesosphere are well known [e.g. Prather, 1981; Pallister and Tuck, 1983]. Although numerous studies have now highlighted observed and modeled peak to peak variations on the order of 5% or more in the middle stratosphere between 30 and 1 hPa [e.g., Sakazaki et al., 2013; Parrish et al., 2014; Schanz et al., 2014a and references therein], adequately resolving the signal on a global scale to account for its effects in data analysis is challenging. Ground-based microwave radiometers have been used to analyze the ozone diurnal cycle at particular locations from the tropics to the northern hemisphere mid- and high latitudes [i.e., Ricaud et al., 1991; Conner et al., 1994; Ogawa et al., 1996; Haefele et al., 2008; Palm et al., 2010; Parrish et al., 2014; Studer et al., 2014; Schranz et al., 2018]. Satellite data provide a more global view of the diurnal cycle. Several satellite missions, including the Upper Atmosphere Research Satellite (UARS) Microwave Limb Sounder

(MLS), the Superconducting Submillimeter-Wave Limb-Emission Sounder (SMILES), and the Sounding of the Atmosphere using Broadband Emission Radiometry (SABER) have made measurements from non-sun-synchronous orbits that capture diurnal variations, but it takes more than a month to sample the full diurnal cycle, over which time the ozone has also undergone seasonal and other geophysical changes.

Thus, it takes averaging over many years or other statistical techniques to isolate the diurnal variations from other sources of variability [e.g., Huang et al., 1997; Huang et al., 2010; Sakazaki et al., 2013]. In addition, these missions do not provide full global coverage.

In this work, we present a climatology of diurnal variability as derived from the NASA Global Modeling and Assimilation Office (GMAO) Goddard Earth Observing System (GEOS) general circulation model

coupled to the NASA Global Modeling Initiative (GMI) chemistry package (GEOS-GMI) [e.g., Oman et al., 2013; Orbe et al., 2017]. The model-based climatology provides a global representation of the diurnal cycle as a function of latitude (5° zonal mean), pressure (~ 1 km equivalent altitude vertical resolution) and season (12 months). Parrish et al. [2014] compared the diurnal cycle in a version of this model to that measured by the microwave radiometer at Mauna Loa and found agreement within 1.5% in most cases

(see Parrish et al., 2014, Figures 8 and 9). Here we expand on those results, analyzing the model diurnal cycle against available measurements over a range of latitudes. As in the Parrish et al. study, most previous observational studies of the diurnal variability in ozone have included simulations from one or more models to support the observed differences, but we are not aware of a model-based climatology of the global diurnal cycle that is easily accessible for use in wide-ranging data applications. In this work,

we do not focus on the chemical and dynamical mechanisms of the ozone diurnal cycle but rather on the validity of the model-derived diurnal climatology as a tool for data analysis. Hereafter we refer to the climatology as GDOC (GEOS-GMI Diurnal Ozone Climatology).

The paper is divided into the following sections: in section 2 we describe the model and the data used in this study; in section 3 we present GDOC and compare its variability to that observed by the SMILES

and the UARS and Aura MLS satellite instruments, as well as to that from previously published observational and model-based studies; in section 4 we explore several example uses of GDOC in data analysis; and finally in section 5 we summarize our results, evaluate the robustness of the diurnal signal in multiple model runs, and detail how to access GDOC.

## 2 Data

## 2.1 GEOS-GMI Output and the Diurnal Ozone Climatology

The diurnal climatology presented in this work is based on output from the NASA GMAO Version 5 GEOS general circulation model, GEOS-5, [Molod et al., 2015] coupled with the NASA Global Modeling

Initiative (GMI) chemistry package [Strahan et al., 2007; Oman et al., 2013; Nielsen et al., 2017], known as GEOS-GMI. Unlike the GEOS Chemistry Climate Model (GEOSCCM) output used in Parrish et al. [2014], which was a free-running model, GEOS-GMI is run in replay mode [Orbe et al., 2017], with dynamics constrained by 3-hourly meteorological fields from the Modern Era Retrospective Analysis for Research and Applications, Version 2 (MERRA-2; Gelaro et al., 2017). The simulation, meant to be

concurrent with measurements from the Stratospheric Aerosol and Gas Experiment (SAGE) III instrument aboard the International Space Station (ISS), is currently available from 2017-2018, and will continue as input fields become available.

Model outputs are instantaneous fields, available every 30 minutes on a 1° by 1° latitude by longitude

spatial grid. The model is run on 72 pressure levels with a model top at 0.01 hPa, and output is interpolated to so-called Z* pressure levels [$Z*=1013.25/10^{(z/16.)}$ hPa for z=0,1,2…80 km] with an approximate pressure-altitude vertical resolution of 1 km (similar to the original model output). Z* pressure levels are often used as a common vertical coordinate when comparing constituent profiles reported (or modeled) on different pressure/altitude surfaces, and is the vertical coordinate used for other climatologies produced

by our group (e.g., the McPeters and Labow [2012] and Labow et al. [2015] profile ozone climatologies). We construct the climatology by averaging two years of output (2017– 2018) as a function of latitude in 5° bins, pressure, month and time of day (local solar time) every 30 minutes. We first average the model output in latitude to reduce the sampling from 1° to 5°. Then at each fixed longitude, latitude, pressure and day, we interpolate in time (at 30-minute resolution) to convert from UTC to local solar time for that

longitude. Note that we sample model output from three consecutive days (in UTC) to get a full local solar time diurnal cycle at each longitude. We then average the diurnal cycles at each longitude to get a daily zonal mean diurnal cycle, and then we average over available days for each month. Finally, for each latitude, level and month, the half-hourly climatological values are normalized to the value at midnight

(11:45-00:15 local time bin) and the final climatology is expressed in terms of variation from midnight. We note that GDOC can be re-normalized to any reference time as is most appropriate for a given analysis.

Uncertainty estimates for GDOC should be based on the standard error of the mean of the model output averaged to construct the climatology. However, with each bin containing 108,000 data points (360 longitude x 5 latitude x 60 days), the standard error of the mean is unrealistically low. The model ozone fields are spatially and temporally correlated, so the true number of independent data points is much lower. To estimate the actual number of independent data points we compute longitudinally lagged correlations at each grid point in a given day and assume that the data points are independent when the lagged correlation drops below a threshold value. Based on this analysis (see Figure S1 and corresponding discussion) we found that there are ~ 12 independent measurements in each daily bin (~ sampling of 30° longitude). Output in each 5° latitude band is also considered to be correlated. Thus, we use n=720 (12 * 60 days) for all computations of GDOC standard error of the mean.

We also use output from the free-running GEOSCCM simulation as presented in Parrish et al. [2014] and from a previous iteration of GEOS-GMI to test the robustness of GDOC to changes in the model formulation (including updates to the input photochemistry and reaction rates) and to different simulation years. Test climatologies from the additional model simulations are representative of different years but are constructed in the same manner.

## 2.2 Ozone Observations

We use ozone observations from multiple data sources to test GDOC in a variety of circumstances. Specifically, we use data from MLS instruments aboard the NASA UARS and Earth Observing Satellite (EOS) Aura platforms; the second generation Solar Backscatter Ultraviolet (SBUV/2) series of instruments since 2000, which include those launched on NOAA satellites 16, 17, 18, and 19; the Ozone Monitoring Profiler Suite (OMPS) Limb Profiler (LP) and Nadir Profiler (NP) instruments aboard the NASA/NOAA Suomi National Polar-orbiting Partnership (S-NPP) satellite; the SMILES instrument

which made measurements from the ISS and the SAGE III instrument currently aboard the ISS (hereafter SAGE III/ISS). Table 1 shows the salient characteristics of the data sets used in this analysis and appropriate references for more information on each instrument.

All high vertical resolution data records except SAGE III/ISS and OMPS LP are reported in pressure coordinates and are first interpolated to Z* pressure levels. SAGE III/ISS and OMPS LP data are reported in altitude coordinates, and MERRA-2 dynamical fields are used to convert between geometric altitude and pressure. OMPS NP and SBUV report ozone as partial column densities (in DU) in pressure layers. Number density and mixing ratio profiles are integrated to give cumulative column densities with pressure, which can be interpolated to re-partition the partial columns to match the SBUV/OMPS NP broad vertical sampling. Monthly climatological averages of satellite data are constructed (with the exception of SMILES and SAGE III/ISS, which are averaged over the entire available time period) in 5° latitude bins. UARS MLS and SMILES are additionally averaged into one-hour time bins. An estimated seasonal cycle is removed from the nine months of SMILES data as outlined in Sakazaki et al. [2013, Figure 3] and the data are not binned by month. Though UARS MLS also samples the diurnal cycle over an extended period, the geophysical variability is largely removed in the 9-year average by month and the error bars capture the remaining variability. In this work, we use UARS MLS data for qualitative comparisons only, and thus do not apply a more rigorous analysis to isolate the diurnal cycle.

## 3 Evaluation of Diurnal Climatology

### 3.1 Characterization of the Diurnal Cycle in GDOC

We first show several examples of GDOC, highlight some of the salient features, and compare generally to past studies. Figure 1 shows GDOC, normalized to the value at midnight, as a function of hour of day and pressure for four latitude bands and months. The ratio is shown with a contour interval of 0.025 (2.5%). The first panel (upper left) shows the climatology for March at 15-20° N. Here the most obvious feature is the low ozone during the day in the lower mesosphere, the well-known mesospheric ozone diurnal cycle [e.g. Pallister and Tuck, 1983]. There is very little, if any, variation in the nighttime values

at these altitudes. Below 1 hPa, there are variations at the sub-5% level. Unlike at higher levels, near 2 hPa the diurnal ozone nighttime values decrease by 2.5% between midnight and dawn, and then vary up and down during the day (see also Figure 2). Results in this latitude band correspond to previous results shown in Parrish et al. [2014], comparing an earlier version of the model to diurnal variations derived from the microwave radiometer at Mauna Loa. Overall, that study showed differences between model and data generally within 1-1.5%. The largest discrepancy was noted in the pre-dawn hours near 2 hPa, where the microwave instrument showed increasing rather than decreasing ozone. However, SMILES data also suggest the ozone is decreasing over this period (Figure 2; Parrish et al., Figure 10). Figure S2 (top panels) show GDOC at 15-20° N for the additional months of January and June. The pre-dawn diurnal ozone decrease is larger in January, as was seen by Parrish et al.

The second panel (upper right) shows results for January at 45-50°N, which can be directly compared to a diurnal climatology developed from the GROMOS microwave radiometer in Bern, Switzerland [Studer et al., 2014, Figure 4a], as well as collocated model output from the Whole Atmosphere Community Climate Model (WACCM) and the Hamburg Model of Neutral and Ionized Atmosphere (HAMMONIA) used in the same study. Compared to the March subtropical climatology in the first panel, the shorter period of daylight hours is evident in the higher latitude January output. GDOC shows a loss of just over 20% at 0.3 hPa, which is somewhat less than that shown by GROMOS or the WACCM and HAMMONIA models, which are closer to 25%. Below about 1.5 hPa, the pattern shifts from daytime low ozone to a pattern of lower ozone in the morning and higher ozone in the afternoon, with variations of more than 5%. GROMOS and the collocated models show a similar shift, though at slightly different altitudes. GDOC agrees more closely with the model output from the GROMOS study, and the authors suggest that the limited vertical resolution of the microwave data might be the cause of the discrepancy [Studer et al., 2014]. This characteristic pattern with higher afternoon ozone in the upper stratosphere diurnal cycle has been widely reported in other observations from ground-based and satellite data [e.g., Haefele et al., 2008, Huang et al., 2008; Sakazaki et al., 2013, Parrish et al., 2014, Schranz et al., 2018]. Using the WACCM model, Schanz et al. [2014a] present a detailed breakdown of the photochemical reactions that contribute to the mid-latitude ozone diurnal cycle at 5 hPa (see also Haefele et al., 2008). Figure S3 shows the

seasonal variability of GDOC at 45-50°N at several altitudes, which matches the higher summertime amplitude model diurnal cycle reported by Studer et al. [2014] and Schanz et al. [2014a].

The lower two panels of Fig. 1 show the diurnal cycle in the northern hemisphere polar summer. The diurnal variability in both the mesosphere and stratosphere is largest near the Arctic Circle (lower left) and decreases nearer the pole (lower right). Near the polar day boundary, the diurnal cycle varies by more than 15% in the stratosphere. This large signal was reported in WACCM output by Schanz et al. [2014a; 2014b]. Recently, one year of microwave radiometer data taken at Ny-Alesund, Spitsbergen, Norway (79° N) showed similar variability with a June peak to peak variation of 5% at 1 hPa (night time ozone higher) and similar amplitude variations but with larger afternoon values at 3 hPa [Schranz et al., 2018]. The authors also included co-located WACCM model results in their analysis, which compared well with the data after accounting for the reduced vertical resolution of the microwave instrument. The high-resolution WACCM output variations are 10% at 1 hPa and 8% at 3 hPa, in very close agreement with the GDOC signal at 75-80° S. Figure S2 (bottom panels) shows the summer polar diurnal cycle in the Southern Hemisphere, which is nearly perfectly symmetric with that in the North.

Figure 2 shows GDOC at 65-70° N as a function of time of day at four pressure levels. Climatological values in March, June, September and December demonstrate the marked variation in the diurnal cycle with season at high latitudes. The summertime (June) diurnal cycle is the largest at all pressure levels. At 0.5 hPa, the square-wave pattern dominates for all seasons, though it is weak in the winter. In the summer, the mesospheric diurnal pattern persists to 1 hPa, while other seasons show a more complicated pattern, with the equinox months showing a secondary peak in the late afternoon. At 3 and 5 hPa, all months except December show an early morning minimum and afternoon maximum. The December diurnal variability is confined to the hours around noon due to limited exposure to sunlight near the polar night boundary. The uncertainty is also greatest in December (winter) at all levels.

A more detailed depiction of the GDOC uncertainty is given in Figure 3. Here we show the uncertainty at noon local solar time (in percent) as a function of month and latitude on four pressure levels. The

uncertainties are very consistent in local solar time, so the noon results are representative of all times. The uncertainties (as defined by the standard error of the mean) are mostly less than 1%. Uncertainties of 1% or greater are highlighted in red, occuring at high latitudes in the winter season of each hemisphere. The largest uncertainties are ~ 2%.

## 5 3.2 Diurnally-Resolved Satellite Data

In Figure 4, we directly compare the general features of GDOC at several pressure levels to those derived from diurnally resolved data from UARS MLS and SMILES satellite-based measurements as well as Aura MLS averages at 1:30am and 1:30pm (black symbols and vertical dotted lines). Specifically, we plot ozone variability as a function of hour of day normalized to the mean daily value for each product. 10 The satellite data tend to be noisy, so we normalize to the daily mean rather than to values at a specific time. Because of their orbital characteristics, both UARS MLS and SMILES have their best coverage within ~ 30° of the equator, so we limit our comparisons to low latitudes. We show results at 15-25°N in Figure 4, but other latitude bands in the tropics are similar. This comparison is qualitative in that we compare the zonal means and we do not attempt to isolate the diurnal cycle in the UARS MLS record 15 beyond simply averaging the data over many years. The deseasonalized SMILES data as derived in Sakazaki et al. [2013] were provided by the authors [T. Sakazaki, personal communication, 2014]. Although the satellite data are noisy from hour to hour, the overall daily variability is accurately represented by GDOC. At 0.5 hPa the mesospheric diurnal pattern prevails, and GDOC captures the amplitude of the day to night ozone differences measured by the satellite instruments. At 1.5 hPa, the 20 pattern is a hybrid of the mesospheric and stratospheric diurnal cycle, with two relative maxima in the early morning and late afternoon, seen also in the SMILES data and to some degree by UARS MLS. Finally at 5 hPa the stratospheric pattern dominates, with measurements and climatology showing the highest daily values in the mid-afternoon. The satellite data agree within ~ 4% on the amplitude of the signal, with GDOC roughly reflecting the average of the satellite data.

### 3.3 Ascending/Descending (Day/Night) Differences

We complete a more rigorous investigation of GDOC by analyzing how well the model reproduces ascending-descending differences in the Aura MLS record. At the equator, Aura MLS makes measurements at 1:30 pm and 1:30 am local solar time, but at other latitudes the exact measurement time varies due to the orbit inclination. Outside of polar latitudes, the ascending measurement is made during daylight hours while the descending measurement is made at night. Hereafter we refer to "day" and "night" rather than ascending and descending. Profiles from GDOC are selected to match the location and measurement local solar time of each MLS profile, and then averaged for direct comparison with MLS day and night averages. For this comparison, when selecting the climatological profiles, we interpolate in time but not in latitude. Figure 5 shows the ratio of daytime to nighttime averages as measured by Aura MLS (top panels) and represented by corresponding profiles from GDOC (bottom panels) as a function of latitude and pressure for two months, June and December.

The day to night ratio in the upper stratosphere, above ~ 1.5 hPa, shows the typical mesospheric diurnal pattern of low ozone in the daytime and high ozone at night [i.e., Pallister and Tuck, 1983]. Below this level the daytime ozone is higher than the nighttime value, but the pattern varies with latitude. As expected, there is little variation between day and night values at high latitudes in polar night [see also Schranz et al., 2018]. In polar day, however, there is a variation of greater than 20% between 5 and 1 hPa near 70° N. Overall GDOC closely matches the spatial pattern and amplitude of the ratios measured by MLS, with agreement generally to within 2%. In the tropics near 1 hPa, we note a local minimum in the day to night ozone ratio in the Aura MLS data. GDOC also shows a local minimum, but the amplitude of this feature is not as pronounced as in the data. It is interesting to note the similarities in the pattern of the diurnal cycle below 30 hPa. However, we do not validate GDOC below 30 hPa because the diurnal variability is small and does not need to be accounted for at these levels.

Figure 6 shows profiles of the day to night ratio from the model and from Aura MLS at 65-70° N and 65-70° S for the months of March, June, September and December. The error bars indicate twice the standard deviation of the Aura MLS profiles averaged from 2004-2018. We show the standard deviation to

highlight the interannual variability of the ratio as measured by Aura MLS. In this case the ratio of the GDOC profiles in the given latitude bin at the ascending and descending time is shown (i.e. GDOC is not explicitly sub-sampled to each MLS profile) and the error bar is twice the root mean square of the two corresponding uncertainty profiles. Though there are some differences between the model simulations

and observations, most notably the small shift in altitude in the June signal at 65-70° N and the offset above 2 hPa in September at 65-70° S, for the most part GDOC reliably reproduces the signal in the observations within 2% or better. Additional profile comparisons of the day to night ratio from GDOC and Aura MLS can be found in Figures S4-S9.

## 4 Example Diurnal Climatology Applications

**4.1 SAGE III/ISS Sunrise Sunset Comparisons**

SAGE III/ISS infers ozone profiles by measuring solar irradiance that has passed through the atmosphere during local sunrise and sunset events. One approach to evaluating these data is by checking the consistency of the measured sunrise and sunset profiles, but care must be taken to account for real diurnal differences between sunrise and sunset. Sakazaki et al. [2015] presented a thorough study of sunrise-

sunset differences from occultation instruments SAGE II, UARS HALOE and ACE-FTS in the tropics between 10° N and 10° S. Their analysis included output from the WACCM Specified Dynamics chemical transport model, and both observations and model indicated an asymmetry between sunrise and sunset measurements in the tropics, with sunrise satellite measurements being larger than those at sunset below ~30 km and above ~55 km.  Figure 7 shows the ratio of mean (2017-2018) SAGE III/ISS sunrise values

to sunset values (SR/SS; red) and that computed from GDOC sub-sampled to match the SAGE III/ISS measurements (blue). Results are shown in three broad latitude bands, and the SAGE III/ISS profiles have been interpolated to pressure levels (using MERRA-2 temperature/pressure data) in this comparison. The SAGE error bars denote twice the standard error of the mean (sem), computed as the root mean square of the sunrise and sunset sem values. The blue error bars for GDOC indicate the variability of the SAGE-

sampled reconstructions (computed in the same way as the SAGE error bars). The overlaid orange error bars (roughly the width of the plotting symbol) represents the model uncertainty, computed as the root

mean square of the model standard deviation profiles at SAGE sampling, divided by the square root of n (=720). Note that the spatial-temporal sampling of profiles is different in the sunrise and sunset averages. By matching the diurnal climatology to each profile, we can represent the impact of the sampling on the ratio, but other geophysical variability that the climatology cannot reproduce may contribute to the measured differences. The SR/SS pattern from GDOC is similar to that reported in Sakazaki et al. [2015] with sunrise profiles greater than sunset profiles (ratio > 1) below ~ 15 hPa (~ 30 km) and above ~ 0.7 hPa (~ 51 km) in the tropics (middle panel). We note that GDOC indicates SR/SS > 1 occurs at 51 km, which is somewhat lower than reported by Sakazaki et al. [2015] in observations (~55 km) and WACCM model results (~ 53 km). At mid-latitudes, the GDOC sunrise/sunset differences are smaller (SR/SS is closer to 1), compared to the tropics, with little difference below 15 hPa and a smaller difference in the upper stratosphere. The GDOC SR/SS pattern is also shifted downward by a few kilometers in the mid-latitudes. The SAGE III/ISS SR/SS ratio generally follows the pattern indicated by GDOC and is within ~ 1% of the GDOC ratio below 2 hPa. Above 2 hPa GDOC and SAGE III/ISS diverge. At these levels, the SAGE III/ISS retrieval does not account for the sharp diurnal gradient in the ozone along the line of sight of the instrument. However, GDOC representations near the terminator may also be biased due to smearing of the diurnal ozone gradient in the monthly average as the terminator time shifts within the month. Also, as noted above, there is some variation between GDOC, WACCM and observations in the SR/SS pattern in the tropics. Nevertheless, these differences suggest potential discrepancies between SAGE III/ISS sunrise and sunset measurements that are currently being explored (R. Damadeo, personal communication, 2019). The purpose of this work is not to evaluate SAGE III/ISS observations but to demonstrate how GDOC can be used in such evaluations.

## 4.2 SAGE III/ISS Comparisons with Other Instruments

As with SAGE III/ISS internal sunrise/sunset comparisons, when evaluating the data relative to independent measurements, the local solar time of the measurements should be taken into account. Occultation instruments measure at local sunrise and sunset while limb and nadir measurements are taken at various times throughout the day, depending on the instrument (see Table 1). In this example, we compare SAGE III/ISS sunrise and sunset profiles to co-located profiles from Aura MLS, OMPS Limb

Profiler (OMPS LP) and OMPS Nadir Profiler (OMPS NP). Both OMPS (LP and NP) and Aura MLS measure at or near 1:30 pm local solar time. In the case of Aura MLS and OMPS LP, co-located profiles are defined as the nearest profile (within 1000 km) to the SAGE III/ISS profile, on the same day, and comparisons are done in altitude. Aura MLS profiles are converted from volume mixing ratio on pressure surfaces to number density on altitude surfaces using co-located MERRA-2 temperature and pressure data. Figure 8 shows mean differences in the 20° - 60° N latitude band between SAGE III/ISS profiles (sunrise and sunset) relative to OMPS LP (upper panel) and Aura MLS (lower panel) before (red) and after (blue) using the diurnal climatology to 'adjust' the SAGE III/ISS profiles to the equivalent measurement time of the correlative data set. Again, our intention is not to do a thorough analysis of the differences but to highlight the influence of the diurnal cycle on such analyses. Near 50 km, the mean differences are reduced by 5% or more when accounting for the diurnal cycle. Similarly, differences are reduced below 44 km, with SAGE III/ISS coming into very good agreement with Aura MLS at these altitudes.

Figure 9 shows comparisons between SAGE III/ISS and OMPS NP profiles in three latitude bands. While OMPS LP is a limb scatter instrument that measures at high vertical resolution, OMPS NP is a nadir backscatter measurement with a broad vertical resolution in the stratosphere. Higher resolution instrument measurements (SAGE III/ISS, MLS, OMPS LP) are often used to help evaluate the lower resolution nadir instruments. This is critical to ensure OMPS NP can continue the 40+ year record of trend quality ozone from the SBUV series of nadir instruments. OMPS NP returns partial column ozone amounts (DU) in pressure layers. Before the SAGE III/ISS sunrise and sunset profiles are averaged, the number density profiles are integrated vertically, giving column densities that are converted to DU and repartitioned into layers that match the OMPS NP vertical resolution. In this case, co-located profiles are the distance-weighted average of all profiles occurring within 1000 km of the SAGE profile on the same day and comparisons are on pressure levels. The top panel shows the mean differences for sunrise-only (yellow) and sunset-only (purple) profiles. The bottom panel shows the same differences after the SAGE III/ISS profiles are converted using GDOC to an equivalent time of 1:30 pm to match the time of the OMPS NP measurements. Note that this comparison is focused lower in the stratosphere than in the previous figure.

As such, the diurnal impacts are smaller. The largest changes are in the 1.0-1.6 and 1.6-2.5 hPa layers, though there are impacts at the 1-2% level in the 6-10 hPa layer and even lower in the tropics. After the diurnal adjustment, the sunrise and sunset biases are closer, and both indicate a shift in the bias above ~ 10 hPa. The remaining pattern of differences is consistent with biases previously reported in the nadir UV

backscatter series of instruments relative to satellite (SAGE II, UARS and Aura MLS) and ground-based (select microwave and lidar) data [i.e. Kramarova et al., 2013; Frith et al., 2017]. Namely, the nadir backscatter instruments tend to have a negative bias below 10 hPa and above 2.5 hPa, and a positive bias near 7 hPa. These examples illustrate how accounting for the diurnal cycle can help to both ascertain the true differences in the profiles and reduce noise in the inter-comparisons.

**4.3 Merging SBUV Ozone Records**

Representing the diurnal cycle is also important when merging multiple ozone data sets to construct a single long-term consistent data record. In this example we consider the SBUV series of nadir-view backscatter instruments, which is used to construct the Merged Ozone Data (MOD) record [Frith et al., 2014; Frith et al. 2017]. The SBUV/2 instruments on NOAA satellites were launched into drifting orbits

such that the measurement time slowly changed over years. In addition, NOAA-17 SBUV/2 was launched into a late morning orbit, while the others were in early afternoon orbits, contributing to differences of several hours in overlapping measurements between instruments. Similarly, NOAA-16, though launched into an afternoon orbit, drifted such that measurements after 2012 were made in the early morning.

The combination of morning and afternoon orbits and drifting orbits can impart diurnally induced bias, drift and seasonal-scale variation between the SBUV/2 data records. We investigate this by comparing NOAA-16, -17 -18 and -19 to Aura MLS data from 2004-2017. Aura MLS volume mixing ratio profiles are integrated to give column density profiles (converted to DU) which are then repartitioned to match the vertical sampling of the SBUV/2 data. Figure 10 shows the 4-6.4 hPa layer ozone difference time

series at 10-15° S. The top panel shows the original differences between each SBUV/2 instrument and Aura MLS, and the bottom panel shows the differences after each SBUV measurement has been adjusted using GDOC to the Aura measurement time. Aura MLS is used as a transfer standard and does not change.

Here the primary impact of the diurnal cycle correction is to reduce the bias between the instruments. At the same latitude band but in the 2.5-4-hPa layer, shown in Figure 11, there are clear drifts over portions of the SBUV records relative to MLS that are largely removed after accounting for the diurnal cycle. Though in this case relative biases between the instruments remain, accounting for a consistent bias in a merged record is much easier than accounting for short-term drifts. Finally, Figure 12 shows the effect of the seasonal variation in the diurnal cycle at higher latitudes (see Figure 5 and Figure S3). Here the SBUV instruments all show a seasonal cycle relative to Aura MLS, but after adjusting for the diurnal cycle the individual SBUV instrument seasonal cycles are in much better agreement relative to MLS. These varied effects can be understood by considering the diurnal cycle in each example, as shown in Figure S10. The SBUV/2 records shown in Figures 10-12 vary in measurement time from 2 to 4 pm and from 8 to 10 am. At 10-15° S at 5 hPa there is a difference in the diurnal cycle from morning to afternoon, but little change between 8 and 10 am or between 2 and 4 pm. However at 3 hPa there is a continuous gradient in ozone as a function of hour from 8 am to 4 pm. Thus, there is not only a bias between morning and afternoon measurements, but also a drift is induced as SBUV measurements shift earlier or later in time between the hours of 8 to 10 am and 2 to 4 pm. Finally, at 50-55°S at 7 hPa there is no diurnal signal in June-July-August but there is a 5% variation between morning and afternoon ozone in December-January-February, leading to diurnally induced seasonal differences between instruments.

## 5 Summary and Conclusions

In this paper, we present a global climatology of the ozone diurnal cycle based on the NASA GEOS-GMI chemistry model. The climatology provides ozone values every 30 minutes during the day, expressed as ratios to the value at midnight (though it can be renormalized relative to other times). It varies as a function of latitude, pressure, and month, with a latitude resolution of 5° and a vertical resolution of ~ 1 km equivalent pressure altitude. Previous studies of diurnal ozone observations often include co-located model results for comparison, but as far as the authors are aware, this is the first easily accessible model-based climatology to be made available for general data analysis purposes. A model-based climatology is useful because no observational data source provides a full representation of the ozone diurnal cycle. However, this fact also makes the model output difficult to validate. Here we compare the climatology to

time-resolved satellite-based data from UARS MLS and SMILES, and compare the day to night climatological ratios to those derived from Aura MLS measurements. We also compare the climatology to previously published results including model analyses and diurnally resolved data from ground-based microwave radiometers. The GEOS-GMI diurnal climatology compares well with all sources; the most quantitative comparison against Aura MLS daytime to nighttime profiles ratios shows agreement typically within 2%.

The diurnal climatology exhibits the largest variability during summer near the polar day boundary (65-70°), as reported previously by Schanz et al. [2014a, 2014b] based on WACCM model output. This is also supported by ratios of daytime to nighttime ozone profiles from Aura MLS. The hourly ozone variation shifts from a mesospheric pattern of low ozone during the day and high ozone at night to a stratospheric pattern of low ozone in the morning and high ozone in the afternoon. However, the amplitude of the signals and the altitude of the transition vary significantly with season, leading to very complicated diurnal patterns that are not easily characterized in data inter-comparisons.

In this work, we do not focus on the chemical and dynamical mechanisms of the diurnal cycle but rather on the validity of the model-derived diurnal climatology as a tool for data analysis. We present a series of examples that highlights the usefulness of the climatology in data analysis as well as demonstrates the consistency between the observed and predicted ozone variations. We represent the uncertainty of the climatological mean values as two times the standard error of the mean of the bin averages, assuming n=720 independent measurements in each bin. This gives error bars that are 2% or less.

The comparisons presented here give us confidence in the climatology, but we also need to consider potential sources of uncertainty. Systematic changes in the diurnal cycle over a month or year-to-year will be smoothed within the climatology. The Aura MLS ASC/DSC ratios (Figures 6 and S4-9) do not suggest significant interannual variability in the large-scale diurnal structure. To further quantify this, we compare GDOC derived using just 2017 model output to that derived using just 2018 model output, as shown in Figure S11. Below 5 hPa the differences are generally less than 1%. At higher levels, there are

sporadic instances of larger differences (3-5%) in the tropics and at higher latitudes but overall, differences remain small. As more years of model output become available, we will be able to better characterize interannual variability in the model. Similarly, true day-to-day or longitudinal variability in the diurnal cycle will be smoothed out in the zonal average over the month. We find varying degrees of

both day-to-day and longitudinal variability in the model diurnal cycles, and this is a subject of ongoing analysis, but characterizing these sources of variability is beyond the scope of this manuscript. Care should be taken when reconstructing daily values using the monthly GDOC, especially near the terminator in the upper stratosphere, where the ozone gradient is sharp and varies in time over the month.

A final source of uncertainty is potential model error. The climatology is normalized, so the only relevant error is representation of the diurnal cycle. To further test the stability of the model diurnal cycle, we consider several different simulations using iterative versions of the model and/or simulations of different years, and compare the diurnal cycle derived from each simulation. Figure S12 shows the December day-night ratios from diurnal climatologies constructed from four separate simulations. The overall patterns

from all the simulations are very similar, suggesting that the representation of the diurnal cycle within the model is well established. This does not preclude a model issue that is present in all model versions. Ideally, as the model is used more in data analyses, such studies will also provide feedback to the modeling team.

We recommend using GDOC primarily for monthly zonal mean analyses in the pressure range from 30 to 0.3 hPa. Comparisons against the various satellite measurements presented in this study suggest that the climatology captures diurnal variations to well within 5% in most cases. For applications that require accurate knowledge of high temporal and spatial resolution changes in ozone we advise using the original model output (see *Data Availability*).

*Data Availability.*

The GEOS-GMI diurnal ozone climatology is stored as a NetCDF file and is available for download on

our local NASA Goddard Code 614 TOMS access site https://acd-

ext.gsfc.nasa.gov/anonftp/toms/GDOC_diurnal/ (last access February 19, 2020) under subdirectory GDOC_diurnal. Also available from this site are the SBUV/2 data (subdirectory sbuv) and OMPS NP data (subdirectory omps_np). These data are also accessible via links from the Merged Ozone Dataset (MOD) website at https://acd-ext.gsfc.nasa.gov/Data_services/merged/instruments.html (last access

February 19, 2020). OMPS LP and NP data and UARS and Aura MLS data are archived at the NASA Goddard Earth Sciences Data and Information Services Center (GES-DISC) (https://disc.gsfc.nasa.gov, last access August 20, 2019). SAGE III/ISS are available at the NASA Langley Atmospheric Science Data Center (ASDC) (https://eosweb.larc.nasa.gov/project/sageiii-iss/sageiii-iss_table, last access August 20, 2019). SMILES data are available from the Data Archives and Transmission System (DARTS)

(http://darts.jaxa.jp/stp/smiles/, last access August 20, 2019). The Mauna Loa hourly resolved microwave data are available by request (A. Parrish, parrish@astro.umass.edu). Additional model output from the current GEOS-GMI simulation is available for collaborators by request (L. D. Oman, luke.d.oman@nasa.gov).

*Author Contributions.*

S. M. Frith conducted the primary analysis including constructing the GEOS-GMI diurnal ozone climatology and applying the climatology to various data analysis tasks. L. D. Oman formulated and ran the model simulations and provided guidance interpreting the model output. N. A. Kramarova provided analysis of OMPS LP and SAGE III/ISS data. G. J. Labow contributed to Aura MLS and SBUV

measurement analysis. P.K. Bhartia conceived the original idea for this work and oversaw its

development, and R. D. McPeters provided funding support and project management. S. M. Frith prepared the manuscript with significant contributions from all authors.

*Competing Interests.*

The authors declare that they have no conflict of interest.

*Acknowledgements.*

S. M. Frith is supported under NASA WBS 479717 (Long Term Measurement of Ozone). Model simulations are supported by the SAGE III/ISS Science Team and NASA MAP programs and the high-

performance computing resources were provided by the NASA Center for Climate Simulation (NCCS). The authors thank R. Stolarski for his helpful comments on the manuscript. We also thank the various instrument teams for providing the data used in this study, particularly those responsible for SAGE III/ISS, Aura MLS, OMPS and SBUV.

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

**Table 1. Ozone Observations and Corresponding Measurement Times.**

| Instrument | Measurement Time at Equator | Period of Data (years) | Reference |
|---|---|---|---|
| Aura MLS (v4.2) | 1:30pm; 1:30am | 2004-2018 | Froidevaux et al., 2008 |
| SAGE III/ISS (aO3) | Local sunrise; Local sunset | 2017-2018 | Chu and Veiga, 1998 |
| OMPS LP (v2.5) OMPS NP (v2.6) | 1:30pm | 2012-2018 | LP: Kramarova et al., 2018 NP: McPeters et al., 2019 |
| SBUV/2 (v8.6) ascending profiles NOAA-16, NOAA-18, NOAA-19 | Orbit drifts slowly between 2pm and 4pm | NOAA-16: 2000-2007 NOAA-18: 2005-2012 NOAA-19: 2009-2018 | McPeters et al., 2013 Bhartia et al., 2013 |
| SBUV/2 (v8.6) descending profiles NOAA-16, NOAA-17 | Orbit drifts slowly between 8am and 10am | NOAA-16: 2012-2014 NOAA-17: 2005-2011 | McPeters et al., 2013 Bhartia et al., 2013 |
| UARS MLS (v5) | Complete cycle 36 days | 1991-1999 | Livesey et al., 2003 |
| SMILES (v2.4) | Complete cycle 30 days | Oct 2009-Apr 2010 | Kasai et al., 2013 |

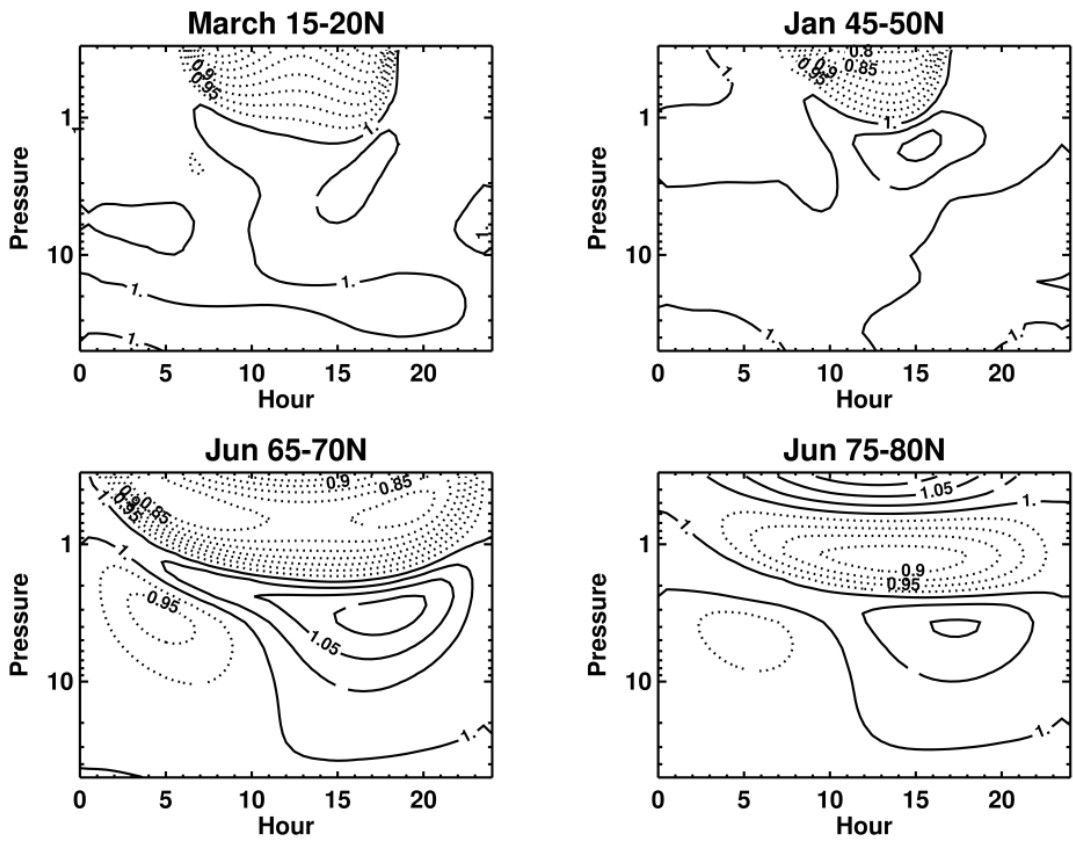

**Figure 1. Contour plot of the GEOS-GMI diurnal ozone climatology (GDOC) normalized to the midnight value as a function of hour and pressure for March at 15-20° N (top left); January at 45-50° N (top right); June at 65-70° N (bottom left); and June at 75-80° N (bottom right). The contour interval is 0.025 (2.5%). The climatology is shown at levels from 50 hPa to 0.3 hPa.**

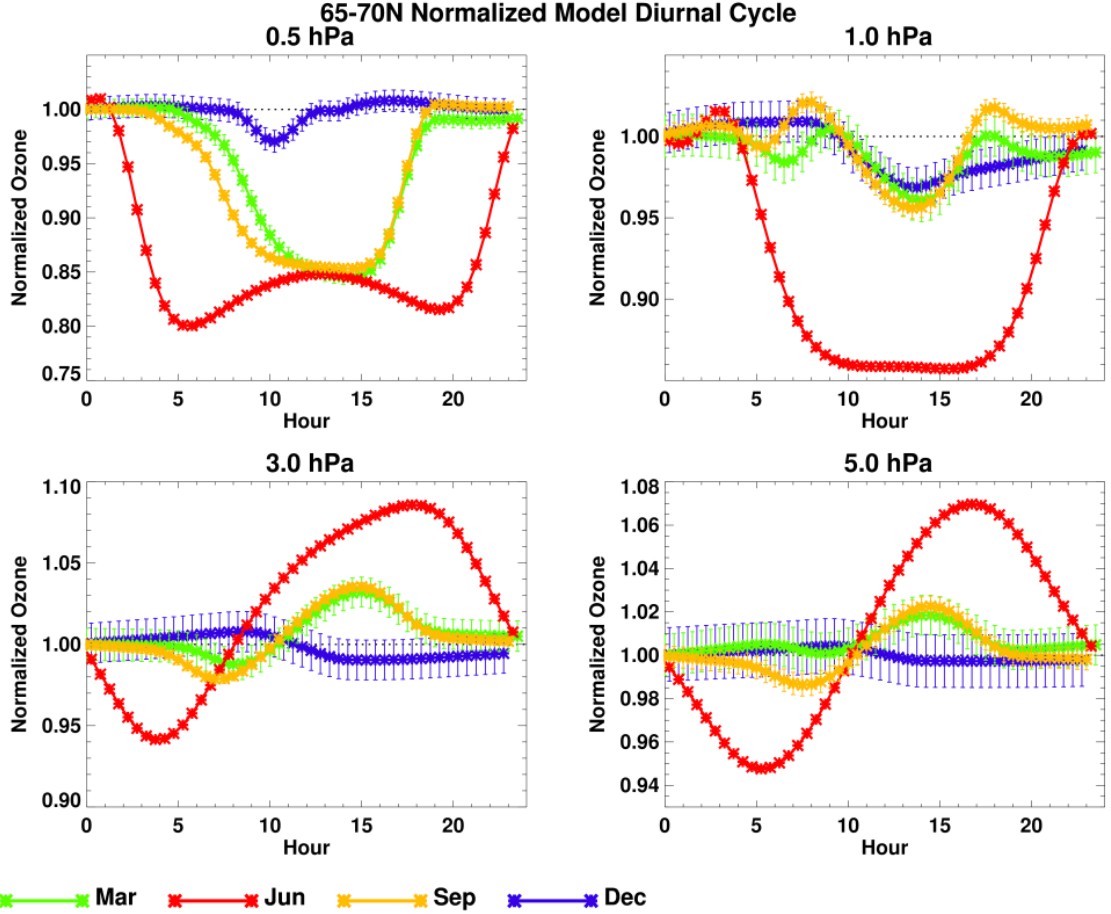

**Figure 2. GDOC at 65-70° N as a function of season on four pressure levels: 0.5 hPa (top left); 1 hPa (top right); and 5 hPa (bottom left); and 5 hPa (bottom right). Seasons are represented by monthly output in March, June, September and December. The diurnal signal is plotted as a function of hour (30-minute resolution) and is normalized to the midnight value. The error bars are 2\*standard error of the mean, as described in the text. The model uncertainty is largest in winter, when the day to day and longitudinal variability of model ozone is greatest.**

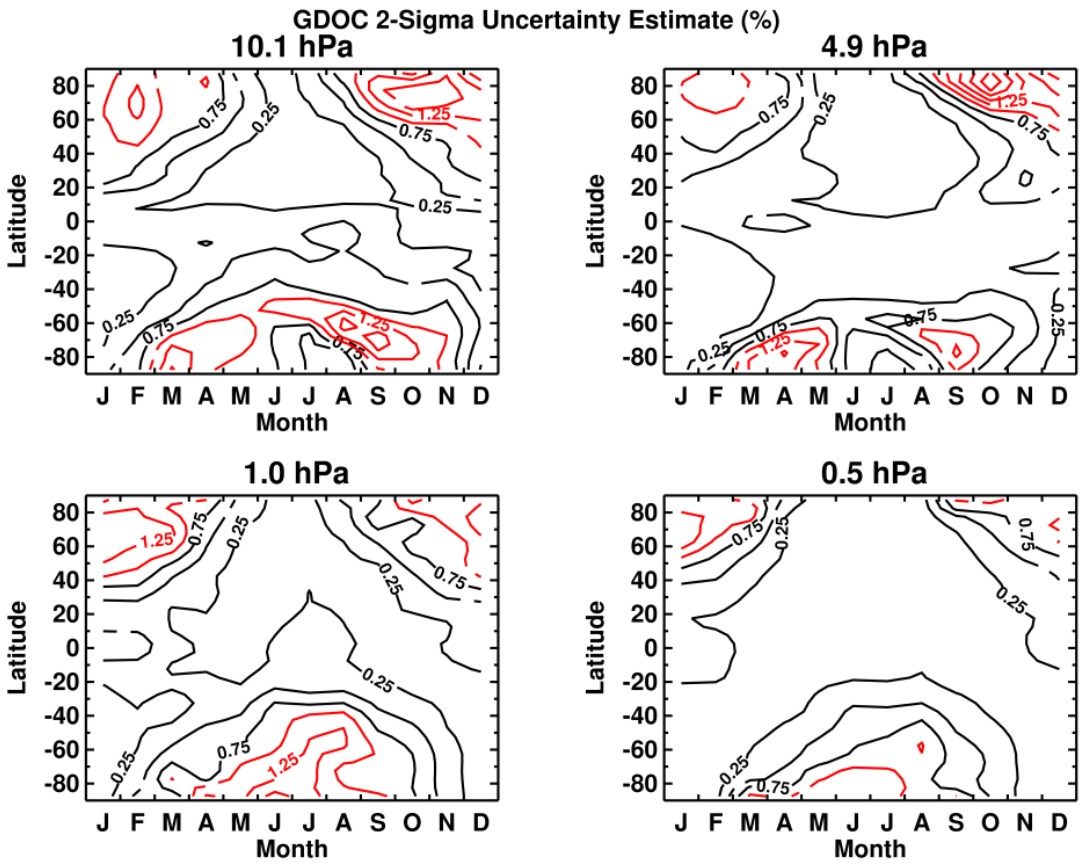

**Figure 3. GDOC uncertainty estimates at noon local solar time, plotted as a function of month and latitude on four pressure levels: 10.1 hPa (top left); 4.9 hPa (top right); 1.0 hPa (bottom left); and 0.5 hPa (bottom right). The uncertainty is defined as the standard error of the mean in each bin, computed assuming 720 independent data points per bin. Contours of 1% and greater are highlighted in red.**

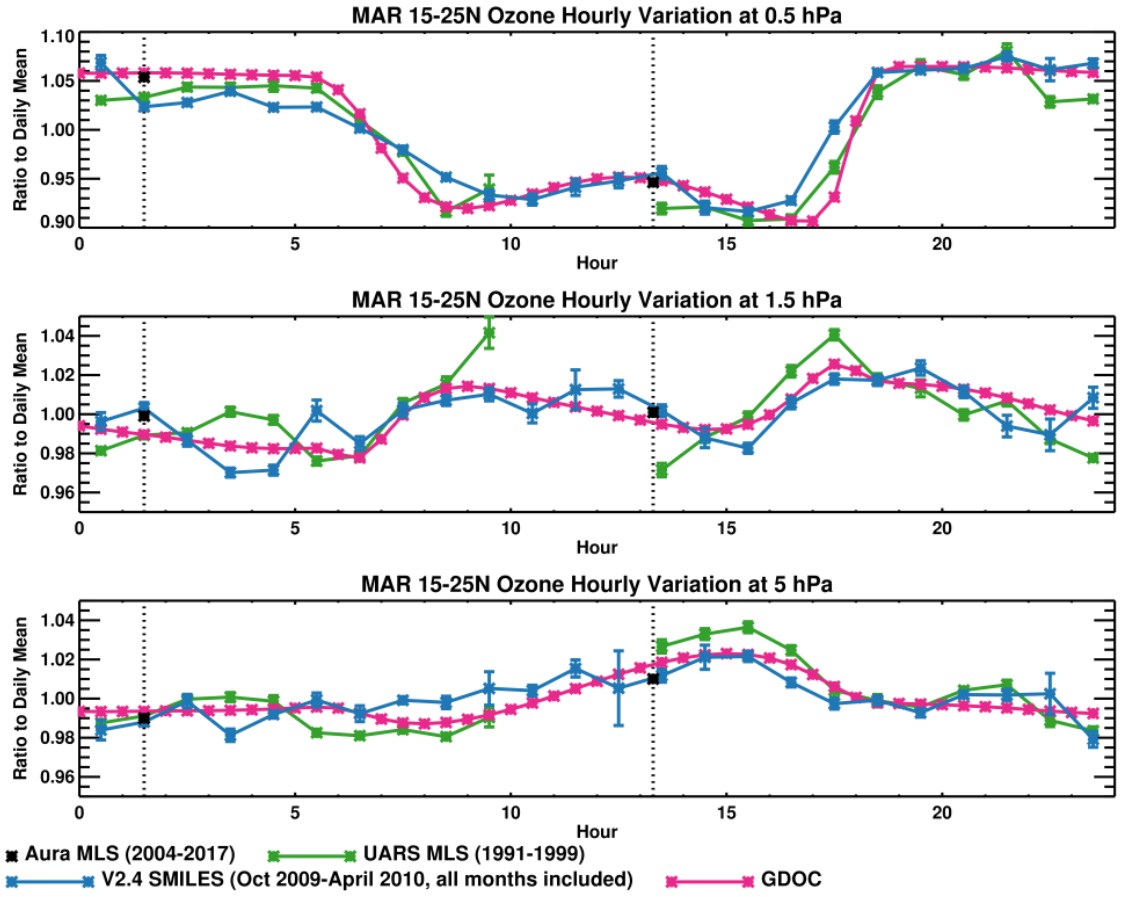

**Figure 4. Diurnal variations as derived from SMILES (blue), UARS MLS (green) and Aura MLS (black symbols), compared to GDOC (red), plotted as a function of hour at three pressure levels: 0.5 hPa (top), 1.5 hPa (middle panel), and 5 hPa (bottom panel). Each product is normalized by its daily mean value, and the ratio is plotted. The black dotted lines indicate the two daily Aura MLS measurement times. UARS MLS means from 10am-1pm are not computed due to limited sampling. The error bars are 2* standard error of the mean. For the model and most satellite averages, this error is smaller than the symbol thickness.**

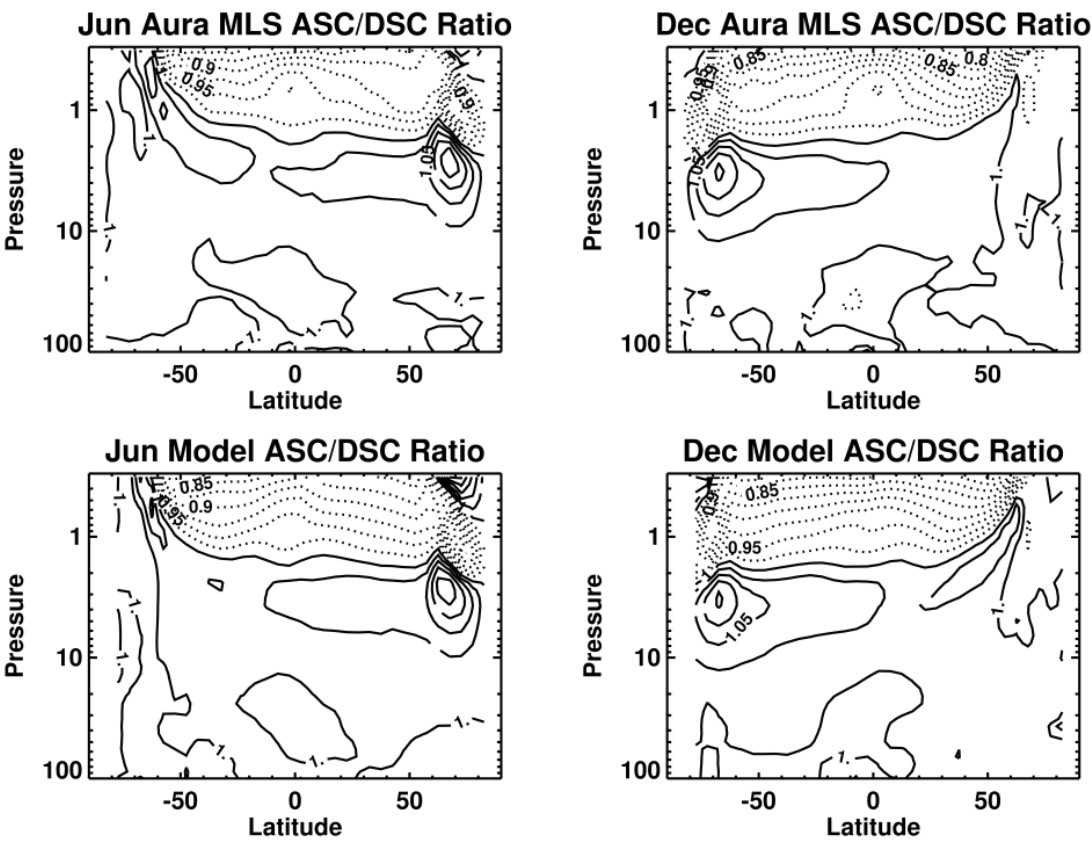

**Figure 5. Aura MLS (top) and GDOC (bottom) average ratio of ascending (day at most latitudes) to descending (night at most latitudes) average ozone in June (left) and December (right) as a function of latitude and pressure from 100 hPa to 0.3 hPa. Contour interval is 0.025 (2.5%). GDOC is sampled at Aura MLS measurement times.**

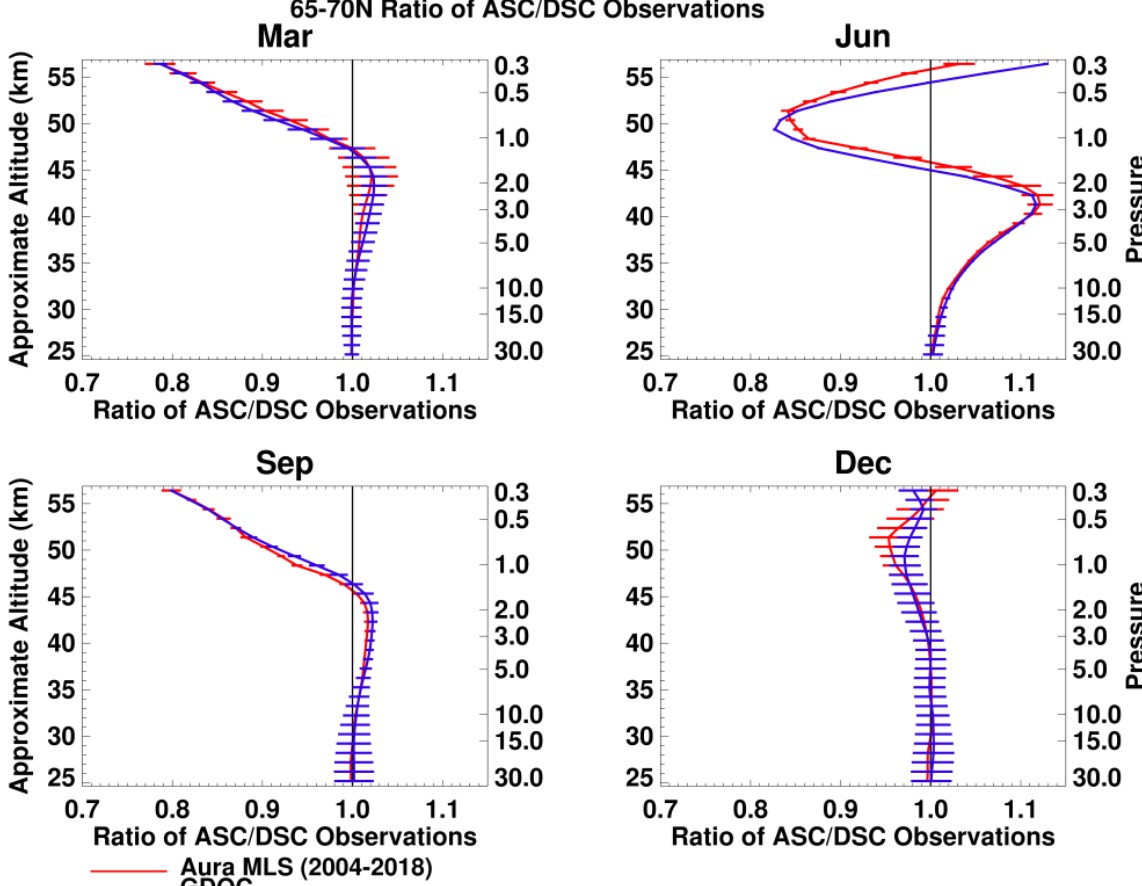

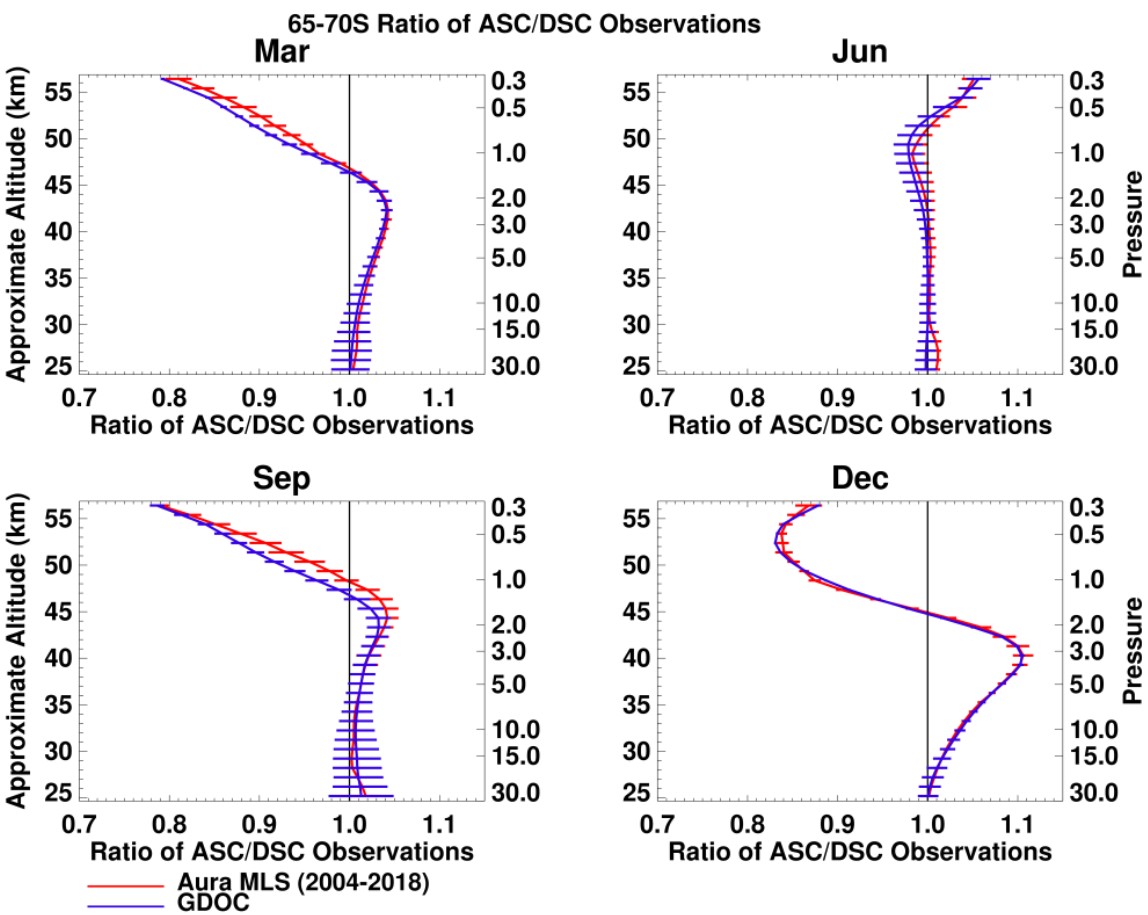

**Figure 6. Profile of mean ratio of ascending (day at most latitudes) to descending (night at most latitudes) measurements at 65-70° N (top four panels) and 65-70° S (bottom four panels) from Aura MLS (2004-2018) and GDOC sub-sampled at Aura MLS profile locations/times. Four panels show results for March, June, September and December. Aura MLS error bars indicate the two-sigma standard deviation of Aura MLS ascending/descending ratio profiles from year to year. We show the standard deviation to highlight the interannual variability of the ratio observed by Aura MLS. The model error bars are 2\* standard error of the mean, as described in the text.**

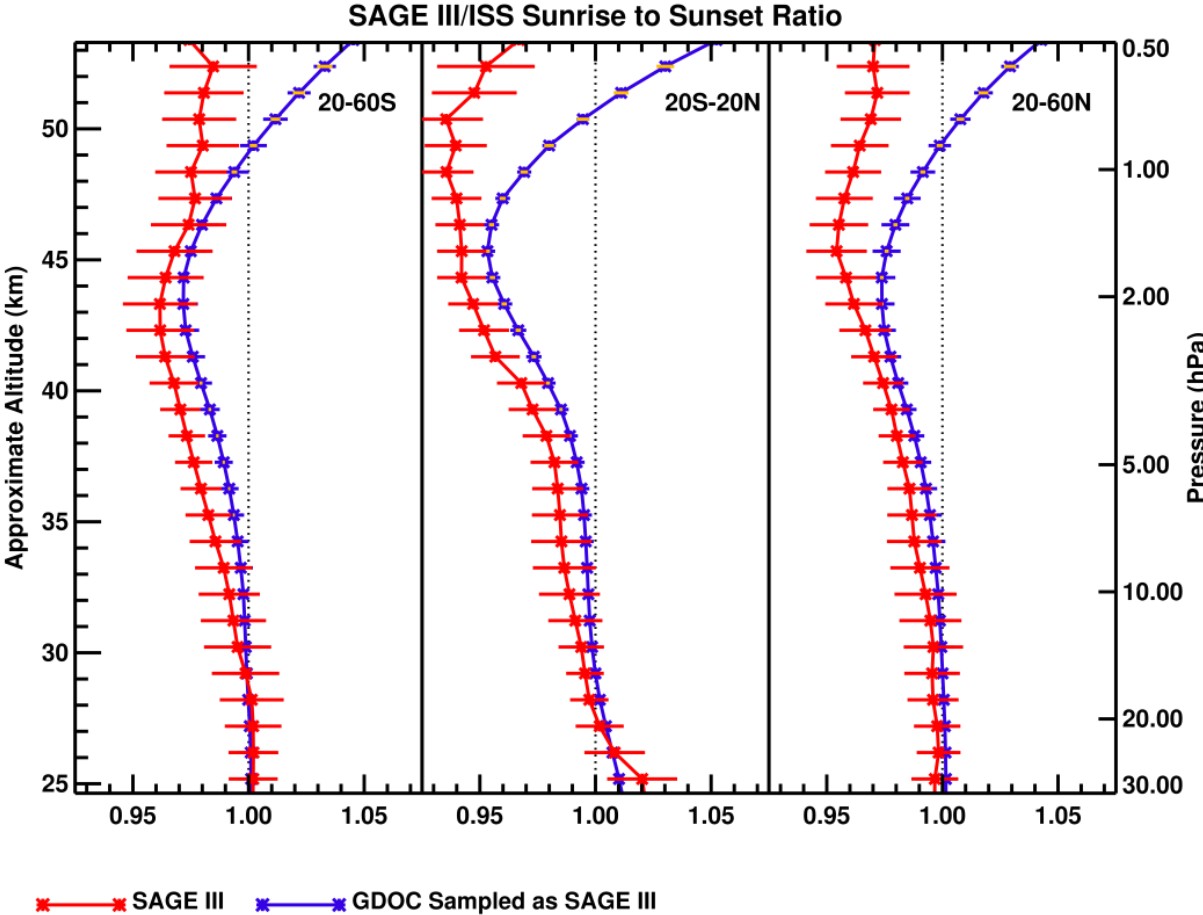

**Figure 7.** Ratio of mean sunrise to mean sunset ozone values from the SAGE III/ISS (red) and from GDOC (blue) sampled at SAGE III/ISS profile locations/times from 2017-2018. Ratios are shown averaged in broad latitude bands: 20-60° S (left); 20° S to 20° N (middle); and 20-60° N (right). The SAGE error bars denote 2*standard error of the mean (sem), computed as the root mean square of the sunrise and sunset sem values. Note that SAGE III measurements are such that the spatial and time sampling are different for the sunrise and sunset mean profiles. The blue error bars for GDOC indicate the variability of the SAGE-sampled reconstructions (computed the same way as SAGE sem). The overlaid orange error bars (roughly the width of the plotting symbol) represents the model uncertainty, computed as the root mean square of the model standard deviation profiles at SAGE sampling, divided by the square root of n (=720).

# SAGE III/ISS – OMPS LP
# SAGE III/ISS-Aura MLS

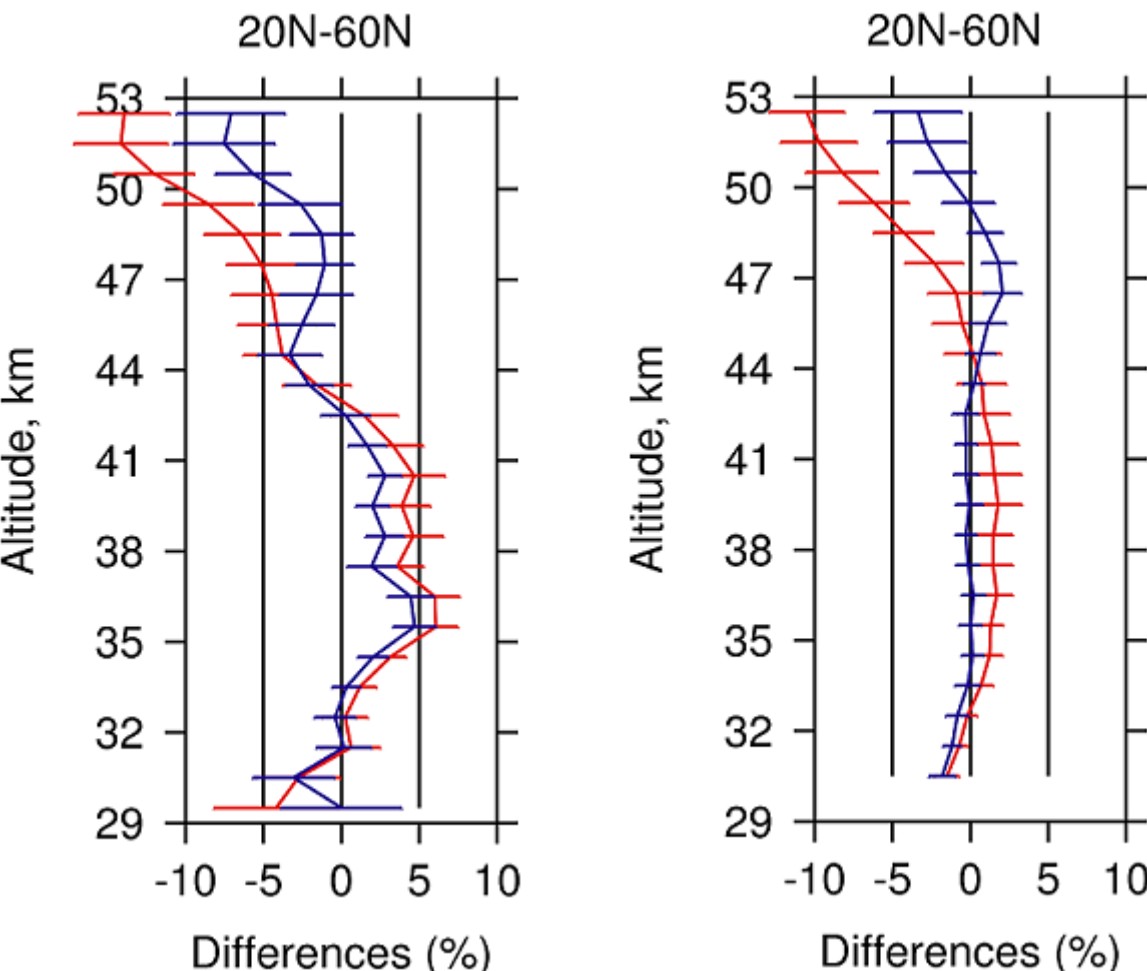

**Figure 8. Profile of mean differences between SAGE III/ISS and OMPS Limb Profiler (left) and Aura MLS (right, daytime measurements only) averaged from 20° N to 60° N, expressed as percent difference as a function of altitude (km). Sunrise and sunset profiles are included in the mean difference. The red curve shows the original mean difference, while the blue curve shows the same comparison after using GDOC to adjust the SAGE profiles to an equivalent measurement time of 1:30pm to correspond to OMPS and Aura measurements. The error bars are the standard deviation (1-sigma; the standard error of the mean is smaller than the line width).**

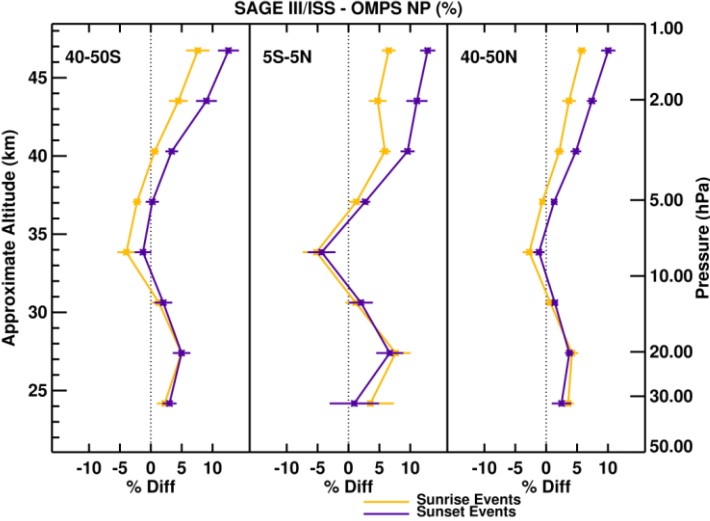

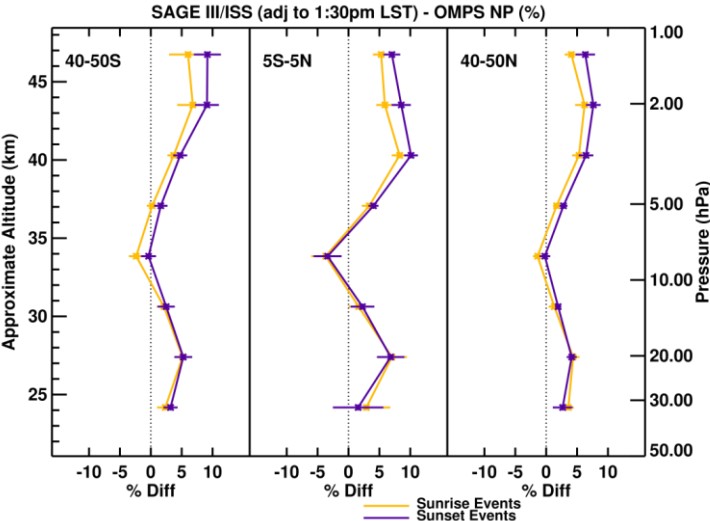

**Figure 9. Profile of mean differences between SAGE III/ISS and OMPS Nadir Profiler (percent difference) as a function of pressure (hPa) separated by SAGE III/ISS sunrise and sunset profiles. Top panel shows original differences and bottom panel shows differences after the SAGE III/ISS profiles have been adjusted to the equivalent measurement time of the OMPS NP profiles. The error bars represent 2*standard error of the mean, based on the month to month variability only.**

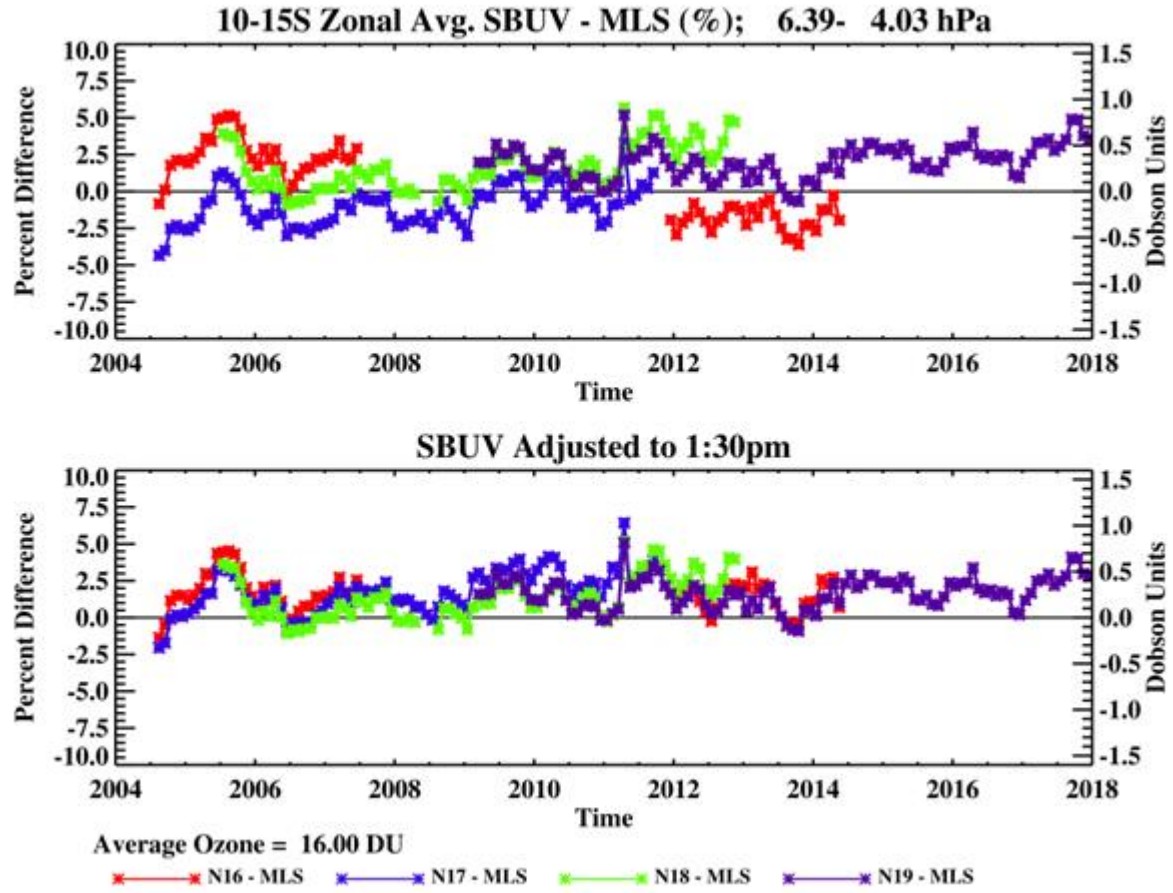

**Figure 10. Time series of NOAA-16 through NOAA-19 SBUV zonal mean data relative to Aura MLS from 2004-2018 in the 10-15°S latitude band and 6-4 hPa pressure layer. Top panel shows original differences and bottom panel shows differences after individual SBUV instruments have been adjusted to a common time of 1:30pm, to coincide with the Aura MLS measurement time. Monthly zonal means of both SBUV and MLS are well sampled such that the uncertainty of 2 * standard error of the mean is smaller than the plot symbols.**

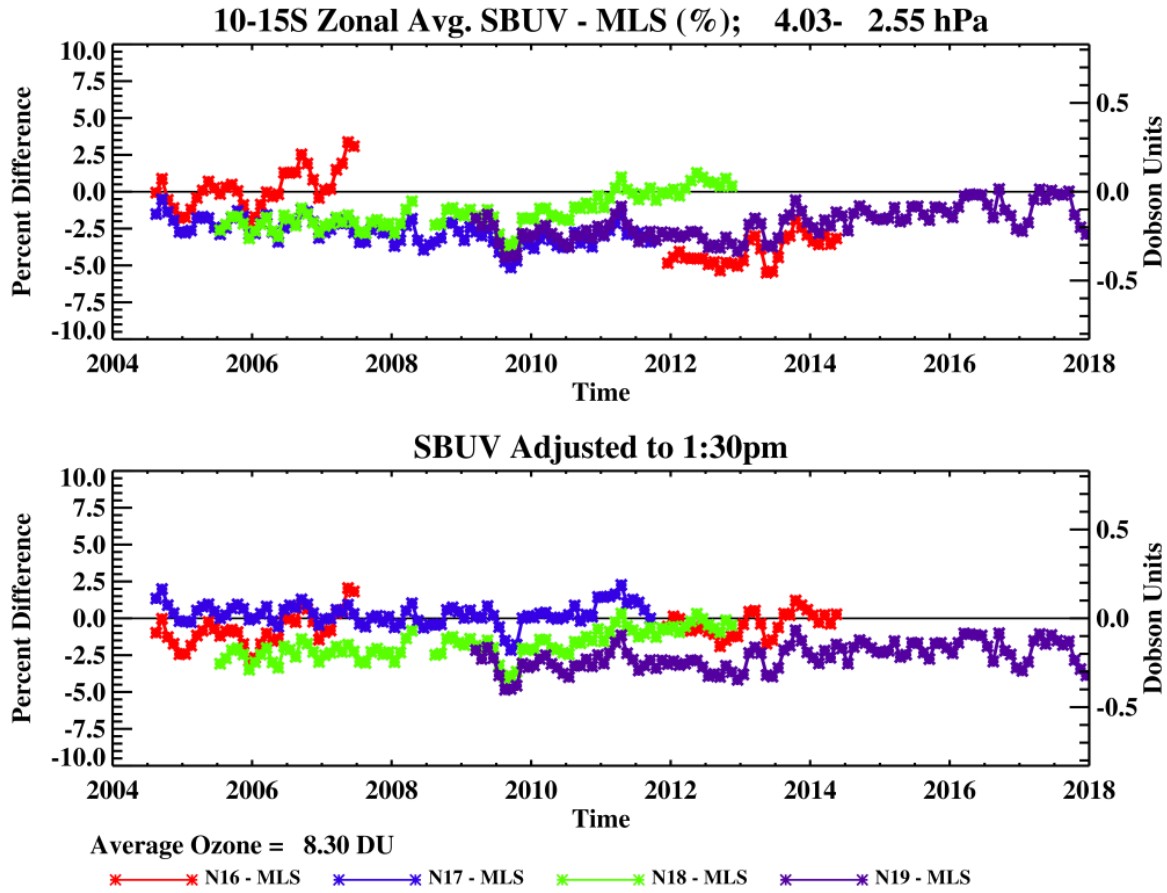

**Figure 11. Same as Figure 10 but for 10-15° S latitude band at 4-2.5 hPa layer.**

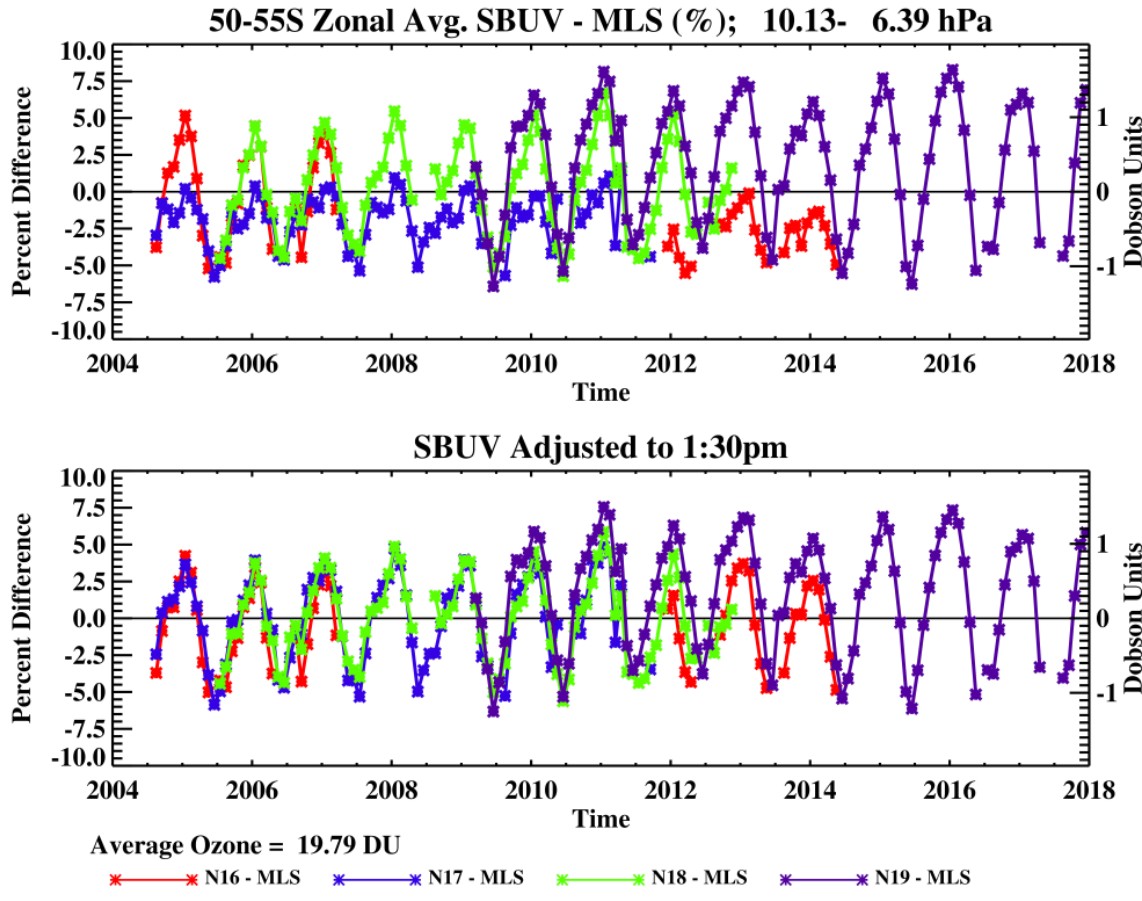

**Figure 12. Same as Figure 11 but for 50-55° S latitude band at 10-6.4 hPa layer.**