# Peer review of "Model-based Climatology of Diurnal Variability in Stratospheric Ozone as a Data Analysis Tool"

_Atmospheric Measurement Techniques, 2019_

## Referee Comment (RC1) · Anonymous Referee #2 · 24 Oct 2019

Review of:

Model-based Climatology of Diurnal Variability in Stratospheric Ozone as a Data Analysis ToolStacey M. Frith1, Pawan K. Bhartia2, Luke D. Oman2, Natalya A. Kramarova2, Richard D. McPeters2, Gordon J. Labow

The study is very detailed, and the results are convincing and new. For the first time, the authors demonstrate a feasible way how the effects of the diurnal ozone cycle in satellite and ground observations can be considered and partly removed. Thus, the article is of high interest for the readers of AMT. Future application of a related analysis to other diurnal cycles in other atmospheric parameters might be possible.

I only found minor corrections which are listed below, and I have one question: I would

be interested in the dependence of the diurnal cycle on longitude. Did you investigate if topography, convection or land-sea contrast have an influence on the diurnal cycle in the simulation data? Maybe you can add 1-2 sentences about this topic to your article.

p.1, line 15 what is the meaning of GEOS-GMI? p.2, line 4 Rowland instead of Roland p.2, line 27 plural? Satellite data provide …. p.4, line 15 0.01 hPa instead of .01 hPa p.4, line 20 please inform how the midnight value is defined, e.g., 23:00-1:00 p.8, line 10 why did you change to the daily mean as reference? p.8, line 19 … measured by the satellite instruments. p.11, line 3 line of sight? p. 14, line 2 … because no observational data source … ? p. 14, line 14 The sentence is not so clear. Perhaps "transits" instead of "transition"?

---

## Referee Comment (RC2) · Anonymous Referee #1 · 6 Nov 2019

This manuscript presents a model-based climatology of diurnal ozone variations in the stratosphere (50-0.5 hPa) based on the NASA GEOS-GMI chemistry model. This climatology is of significant utility for observational data inter-comparisons and merging activities as it allows to correct for diurnal sampling biases in ozone records. This is a topic of high relevance for readers of AMT. The paper is well written and covers all the relevant details and citations. I recommend publication after addressing my comments below, most of them being minor.

General comment:

Overall I'm missing a more quantitative discussion on uncertainties and limitations when using the diurnal climatology in different applications. I see three potential sources of uncertainty: (i) model errors, (ii) unresolved inter-annual variability, and

(iii) climatology discretization. While (i) is very difficult to quantify, (ii) and (iii) could be assessed in a straight forward manner.

The influence of inter annual variability is already discussed in a qualitative way (e.g. Fig S10, differences between 2017 and 2018 outputs) but could be extended to include quantitative estimates.

Regarding (iii), a mayor source of uncertainty could be the relative broad temporal resolution of the climatology (monthly) which may introduce systematic deviations close to the terminator, particularly in the polar regions and at upper stratospheric levels (and above) where photochemistry is relatively fast (intra-month terminator variations are not resolved by the monthly climatology). These errors could be evaluated by e.g. applying the climatology-based diurnal correction to the 0.5-hourly resolved model output itself. Further, an upper vertical limit for a "safe use" of the climatology, would be helpful, particularly when considering that the climatology is provided up to ∼80 km (0.01 hPa).

Specific comments:

p1 l21: "polar summer boundary" -> consider to rephrase to "polar day terminator"

p4 l14-15: The reason for the vertical interpolation is not clear. Why switching to a different vertical grid if the climatology is provided on pressure levels and the interpolated levels have a similar vertical resolution as the original pressure levels? Further, Z* and pr are not defined.

p4 l17-20: I guess that local solar time (LST) is meant with "time of day". Can you provide some more details on how the local time binning has been performed? Was the model output at different longitudes (but fixed UT) resampled to local time or was the local time (at fixed longitude) sampled from the output at different UT (and finally zonally averaged)? This question is relevant since the former option (while in principle allowing for better local time resolution) may introduce aliasing effects by e.g. stationary planetary waves while the latter option is much less sensitive to such aliasing effects.

p4 l25-29: Can you quantify the agreement of the climatologies in Figure S10? The difference of the climatologies for different years could provide a good estimate of the uncertainty range caused by intra-annual variability.

p9ff (Day Night Differences): Apart of Aura/MLS there are also other ozone-observing instruments on sun-synchronous platforms, some of them having different equator crossing LSTs compared to MLS. MIPAS on ENVISAT, for example, took sun-synchronous measurements at 10 am - 10 pm equator crossing LST, in principle allowing to extend the validation of the diurnal climatology by means of observed day-night differences to different LSTs.

p11 l1-4: A possible reason for the divergence between GDOC and SAGE-III above 2 hPa could also be related to the limitations of the monthly-resolved diurnal climatology: sunset (SS) and sunrise (SR) times are spread over a certain LST range in the monthly climatology, resulting in an artificial smearing of the diurnal gradient at SS and SR and hence in reduced SR/SS ratios.

p15 l4: the webpage is not accessible.

―――――――――――――――

---

## Referee Comment (RC3) · Anonymous Referee #3 · 7 Nov 2019

Referee Report

Model-based Climatology of Diurnal Variability in Stratospheric Ozone as a Data Analysis Tool, by Frith et al.

This manuscript describes the (GEOS-GMI) global model climatology (12 monthly sets) regarding ozone diurnal change as a function of local time for various latitude bins and pressure values. The chosen time step (resolution) is a half-hour. Model values are compared to various data sets, mostly from satellite-based ozone measurements with different spatio-temporal samplings. Most of the comparisons seem to validate the model results, even if there are a few discrepancies that are not completely explained. This model climatology is publicly accessible (or will be), and this offers a useful tool for other investigators, to try to improve certain upper stratospheric and mesospheric

ozone comparisons.

General Comments

The paper is generally well-written, clear enough, and fairly thorough in the set of comparisons that are provided for validation. It does not purport to solve in detail every intercomparison's discrepancies. However, the lack of error bar discussion does raise some concerns, regarding the applicability for users; while the comparisons do indicate that the model provides a good representation of the true diurnal changes for ozone, the small differences that come up in terms of inter-instrument trend comparisons, for example, might still be "explained away" by uncertainties in model-based corrections, even after diurnal adjustments. Other uncertainties involve actual line-of-sight gradient issues, not just for the model results, but also for satellite-based retrievals, in particular, for solar occultation results (for which some attempts have been made to adjust for such gradients, but not as a general rule). These issues are the more difficult aspects, but this does not preclude, in my view, publication of this sort of manuscript. I ask for minor clarifications and some attempts (at least) at a better discussion regarding uncertainties, see my specific comments below. I also provide editorial-type comments, mostly as suggestions or corrections.

Specific Comments

1) One somewhat confusing detail has to do with the normalization time. For example, pg. 4, line 20, and pg. 6, line 4 refer to midnight as a normalization time. The Fig. 1 caption agrees with this description. However, the caption for Fig. 2 refers to 1:30 am as the normalization value, and so does Fig. S9. It would be good to clarify why there are these different normalization times, or if they should be the same. It probably does not matter too much, if different Figures are normalized slightly differently, but I found this confusing, so if something is written in error there, please correct.

2) Error bars are not always described (e.g., for Fig. 3), or justified (e.g., why not use standard error in the mean rather than standard deviation for Fig. 5 and Fig. 7, and

similar Figures in the Supplement?). When using a very large data set (e.g., 2004-2018 Aura MLS data in Fig. 5), the random source of error will basically disappear. As an aside, geophysical variability probably accounts for some of the year-to-year differences; differences in day/night temperature or $H_2O$ ratios, for example, could have some impact on $O_3$ abundances and $O_3$ diurnal change. In the mid- to upper stratosphere, $N_2O$ day/night variability from year-to-year (or month-to-month) could impact ozone and its day/night ratios. Some comments about why the authors chose to use standard deviations rather than errors in the mean would be welcome (is it to try to encompass such geophysical variability, which would be ignored in a standard error minimum type of error representation?). Maybe the standard deviation is indeed a more acceptable way to try to encompass sources of error, but I would welcome a brief comment regarding this point somewhere.

3) In some places, there is a mention of vertical "integration" of MLS data to match the vertical resolution of SBUV. This sort of smoothing is best done via the use of MLS Averaging Kernels (and MLS a priori data), although this can be somewhat tedious. The details are not mentioned here, but probably some indication of the "smoothing" or averaging process should be provided. Is there no smoothing in the Figure 5 results? Maybe errors in this, or omission of this, could lead to differences or discrepancies in the results (?). [It would also make more sense to smooth the MLS data sets for day and night and then calculate the ratios, than to smooth the MLS ratios, not that this is what was done].

4) For Figure 6 in particular, the model could be used, in theory at least, to calculate line-of-sight differences in ozone signal for a solar occultation measurement, using small time steps for such a "ray-tracing" calculation, including height-dependence. Comparisons to a case assuming homogeneous line-of-sight ozone abundances, which is often assumed in retrievals, could be made. In theory, the sign of the differences in this case (model versus observations) could thus be ascertained. The authors could at least expand on this by stating that these comparisons are difficult because of not only the model calculation aspects but also the satellite retrieval aspects (they do mention the model, it seems, but not the satellite retrievals explicitly). It is alright to state that such detailed analyses are needed to better ascertain whether the model and data really disagree, even if the more detailed work is not pursued in this manuscript. Also, I wonder if one would not need finer sampling of the model in local time to match the fast changes at sunrise or sunset...(I am not asking to necessarily carry this research out in detail here).

5) Error bars: I would note that there are no error bars in Figure 6, so either they are too small, or just not calculated (as a standard deviation of the ratios, as done in other Figures), probably the latter. Including such error bars would make sense, however. Also, the error bars in Fig. 8 seem to be indicated by dashed lines, a different format, but please explain these ranges in the caption. Also, in Figure 9, maybe a standard error in the mean values as a function of time here would be more appropriate, but no error bars are shown; some comments regarding this (or actual error bars) would be appreciated as well. I expect that the volume of data used for these comparisons (for each specific month) is large enough to ensure that random errors become negligible.

Editorial-type Comments / Suggestions

- Page 1

L14, add a comma after "this issue".

L16, change "applied in" to "applied to".

- Page 2

L3, decide if use ODSs or ODS (I would follow the WMO Report type of writing, so probably ODSs for plural, elsewhere also)

L6, change "has been" to "have been".

L24, "to analyze the ozone diurnal cycle at ..."

[Figure]

L28, change "Atmospheric" to "Atmosphere".

- Page 3

L3, "non-sun-synchronous"

L6, change "source" to "sources".

L7, I suggest "Also, these missions do not provide full global coverage."

L25, "as well as to that from..."

- Page 4

L19, it seems that "semi-hourly" should replace "hourly" here, since you use 30 minute model time steps.

- Page 5, L7. You mention OMPS NP and OMPS LP. You also later refer to OMPS profile data and mention NP (top of page 12). Please clarify which data set is being used, NP or LP (or both?), as this was not quite clear enough; maybe this mainly requires a change on page 12. If the datasets are used as mentioned (LP for one plot, NP for another), please clarify (briefly) why one should use LP versus NP or vice-versa (what are advantages/disadvantages of NP versus LP?).

- Page 6

L8, add commas "...very little, if any, variation..."

L12, add a comma after "Parrish et al. [2014]".

L15/16. However, SMILES data also suggest that ozone is decreasing..."

L22, add a comma after "Figure 4a]".

L28, either say "variations greater than" or "variations of more than"

- Page 7

L2, authors suggest that the

L8, Delete "Supplemental"

L9/10, matches the higher summertime amplitude model diurnal cycle reported by Studer...

L11, panels of Fig. 1 show the diurnal cycle...

L14, change "greater" to "more".

L18, but with larger afternoon values at 3 hPa

L22, delete "Supplemental" [also, it is a bit strange to refer to S1 after you referred to S2 earlier]

- Page 8

L20, relative maxima.

L22, relatively high ozone value.

- Page 9

L7, add a comma after "this comparison"

L8, shows the ratios of daytime to nighttime averages

L17, amplitude of those in the MLS data, with ratios generally ...

L18, near 1 hPa, we note a local minimum in ...

L19, local minimum

- Page 10

L2, delete "Supplemental"

L23 and L26, (maybe) change middle latitudes to midlatitudes

- Page 11

L3, change "site" to "sight"

L18, add a comma after OMPS NP. also, please state briefly how the conversion for MLS O3 profiles from pressure to altitude is made.

L20, change "show" to "shows"

L24, influence of the diurnal cycle on such analyses

- Page 12

L11, please add a sentence or two describing the "known bias pattern" for nadir UV instruments... Not everyone is familiar with what this means, and readers should not have to try to dig this out from other references (top-level information at least); "bias pattern" versus what? (in general?).

- Page 13

L10, please specify which instrument's results show a larger (or smaller) amplitude, is it MLS or SBUV, since the differences do not provide the reader with this information. One would think that the finer resolution instrument might provide a larger amplitude, although the broader vertical extent of the SBUV views means that this is actually not obvious.

L13, delete "Supplemental"

L24, change ozone levels to ozone values

L25, expressed as ratios to the value at midnight

- Page 14

L11, change "depicts" to "exhibits"

L27, suggesting that the representation...

- Page 21, change Froidevaux to Froidevaux et al.; also change Livesay to Livesey.

- Figure 1: the caption says "30 hPa to 0.3 hPa" but the plots seem to go down to 50 hPa. Please clarify.

- Many of the Figures spell "AURA" rather than "Aura", which is the correct spelling (it is not an acronym), as spelled correctly in most of the manuscript. It would be good to correct the Figures for this. Also, Fig. 4, Fig. 5, and others in the Supplement have Day/Night Ratio as plot titles, but show ASC/DSC (for Fig. 5) in the axis labels... In reality, day and night during polar summer or winter does not make sense, as it is always either day or night, so it is more correct to state ASC/DSC as what is being calculated, if I am not mistaken. If this is true, the Day/Night labels should more properly be written as ASC/DSC, and for consistency with axis labels... At most latitudes, of course, this is the same thing... In Fig. 5 (and others like it) there are confusing y-axis tick marks on the right side; it would be best to delete the altitude tickmarks there. Also, in Fig. 9, the last sentence could be rewritten a bit as "adjusted to *a* common time of 1:30 pm, to coincide with *the* Aura MLS measurement time." Figure S10: change "output" to "outputs" in the last sentence; it would have been nice to indicate what mostly contributes to the differences between the 2017 and 2018 runs (is it geophysical variability in the model, or were there also some sampling differences in how this was calculated, if matching SAGE sampling patterns?).
* * *

---

## Author Comment (AC1) · 20 Feb 2020

This manuscript presents a model-based climatology of diurnal ozone variations in the stratosphere (50-0.5 hPa) based on the NASA GEOS-GMI chemistry model. This climatology is of significant utility for observational data inter-comparisons and merging activities as it allows to correct for diurnal sampling biases in ozone records. This is a topic of high relevance for readers of AMT. The paper is well written and covers all the relevant details and citations. I recommend publication after addressing my comments below, most of them being minor.

** We thank the reviewer for their comments and address each point individually below, as indicated by the bold text. We note that during the review process a model error was

[Figure]

identified and a new simulation was run. We reanalyzed the new output, but found for ozone the differences were very small, and did not warrant producing a new climatology at this time. We will periodically update the climatology and include all model updates at that time.

Overall I'm missing a more quantitative discussion on uncertainties and limitations when using the diurnal climatology in different applications. I see three potential sources of uncertainty: (i) model errors, (ii) unresolved inter-annual variability, and (iii) climatology discretization. While (i) is very difficult to quantify, (ii) and (iii) could be assessed in a straight forward manner. The influence of inter annual variability is already discussed in a qualitative way (e.g. Fig S10, differences between 2017 and 2018 outputs) but could be extended to include quantitative estimates. Regarding (iii), a major source of uncertainty could be the relative broad temporal resolution of the climatology (monthly) which may introduce systematic deviations close to the terminator, particularly in the polar regions and at upper stratospheric levels (and above) where photochemistry is relatively fast (intra-month terminator variations are not resolved by the monthly climatology). These errors could be evaluated by e.g. applying the climatology-based diurnal correction to the 0.5-hourly resolved model output itself.

** We agree, and we have added a section in the summary with a more thorough discussion of the potential model errors. We have also made a good faith effort to include reasonable error bars for the climatology. In doing so we analyzed the variability of the high-resolution data going into the climatology (equivalent to re-sampling the model output from the climatology). The standard deviations are large, over 10% in high latitude winter. The climatology will smooth out sub-scale variability related to the diurnal cycle, but we weren't able to isolate that variability from the overall noise. We found very little difference in the standard deviations from hour to hour, suggesting that the longitudinal variability is dominating the variability due to day to day variations in the terminator time. We added a cautionary note in the summary with regard to using GDOC near the terminator. We have added the figure below showing the direct

differences between the model simulation in 2017 and 2018 as a function of season and latitude at 4 pressure levels. The difference plotted is the max-min difference in local solar time.

Further, an upper vertical limit for a "safe use" of the climatology, would be helpful, particularly when considering that the climatology is provided up toâĹij80 km (0.01 hPa).

** We suggest a safe use range of 30 hPa to 0.3 hPa. This has been added to the text in the same conclusions section, and the actual data set will be truncated.

** Updated Summary Paragraph: In this work we do not focus on the chemical and dynamical mechanisms of the diurnal cycle but rather on the validity of the model-derived diurnal climatology as a tool for data analysis. We present a series of examples that highlights the usefulness of the climatology in data analysis as well as demonstrates the consistency between the observed and predicted ozone variations. These comparisons give us confidence in the climatology, but we also need to consider the potential sources of error. The first measure is the variability of the data going into the climatology. We use the standard error of the mean of the bin averages, assuming n=720 independent measurements in each bin, as the measure of this uncertainty. Uncertainty estimates based on this variability are 2% or less. However, care should be taken when reconstructing daily values using the monthly GDOC, especially near the terminator in the upper stratosphere, where the ozone gradient is sharp and varies in time over the month. True day-to-day or longitudinal variability in the diurnal cycle will be smoothed out in the averaging. Other potential sources of error are interannual variability and model error. In an effort to address both these issues, we consider several different simulations using iterative versions of the model and/or simulations of different years, and compare the diurnal cycle derived from each simulation. Figure S10 shows the December day-night ratios from diurnal climatologies constructed from four separate simulations. The overall patterns from all the simulations are very similar, suggesting that the representation of the diurnal cycle within the model is well estab-
lished. This does not preclude a model error (or more likely incomplete representation of relevant processes) that is present in all model versions. Ideally, as the model is used more in data analyses, such studies will also provide feedback to the modeling team. For more information on year-to-year variability, Figure S11 shows direct differences between GDOC derived from the 2017 and 2018 simulations from the same model. Below 5 hPa the differences are generally less than 1%. At higher levels there are sporadic instances of larger differences (3-5%) in the tropics (also seen in Figure S10) and at higher latitudes. As more years of model output become available, we will be able to better characterize interannual variability in the model. We recommend using GDOC primarily for monthly zonal mean analyses in the pressure range from 30 to 0.3 hPa, and expect the climatology to capture diurnal variations to well within 5% in most cases. For finer resolution studies, GDOC can be used in a first order effort to estimate the impact of the diurnal cycle, to be followed by analyses that are more refined. Users requiring more highly resolved information may contact the authors for access to the original model output.

Specific comments :p1 l21: "polar summer boundary" -> consider to rephrase to "polar day terminator"

** done

p4 l14-15: The reason for the vertical interpolation is not clear. Why switching to a different vertical grid if the climatology is provided on pressure levels and the interpolated levels have a similar vertical resolution as the original pressure levels? Further, Z* and pr are not defined.

** We have clarified this section, pr was meant to be Z* pressure levels. This was done for convenience; we often use Z* coordinates as a common vertical coordinate when comparing data sets. We noted this in the text: " Interpolation to Z* levels is done largely for convenience; Z* pressure levels are often used as a common vertical coordinate when comparing ozone profiles from a variety of instruments reporting on

different altitudes."

p4 l17-20: I guess that local solar time (LST) is meant with "time of day". Can you provide some more details on how the local time binning has been performed? Was the model output at different longitudes (but fixed UT) resampled to local time or was the local time (at fixed longitude) sampled from the output at different UT (and finally zonally averaged)? This question is relevant since the former option (while in principle allowing for better local time resolution) may introduce aliasing effects by e.g. stationary planetary waves while the latter option is much less sensitive to such aliasing effects.

** Yes, we mean local solar time, and that has been added to the text. The binning is done by sampling local time at a fixed longitude from output at different UTC. We have added the following details to the text to better explain how the local time binning is accomplished: We first average the model output in latitude to reduce the sampling from $1°$ to $5°$. Then the binning in local solar time is accomplished by sampling from three consecutive days of half-hourly output. At each fixed longitude, latitude, pressure and day, we construct a 3-day time series (30 minute resolution, centered on the fixed day) of ozone values in UTC time, then integrate in time to convert from UTC to local solar time for that longitude. Thus, there is a diurnal cycle defined at each longitude. We then average the diurnal cycles at each longitude to get a daily zonal mean diurnal cycle, then we average over available days for each month. Finally, for each latitude, level and month, the half-hourly climatological values are normalized to the value at midnight (11:45-00:15 local time bin) and the final climatology is expressed in terms of variation from midnight. We note that GDOC can be re-normalized to any reference time as is most appropriate for a given analysis.

p4 l25-29: Can you quantify the agreement of the climatologies in Figure S10? The difference of the climatologies for different years could provide a good estimate of the uncertainty range caused by intra-annual variability.

** Yes, we have added Figure S11 to the Supplemental, and reference it in the text [see

above]. Figure is attached.

p9ff (Day Night Differences): Apart of Aura/MLS there are also other ozone-observing instruments on sun-synchronous platforms, some of them having different equator crossing LSTs compared to MLS. MIPAS on ENVISAT, for example, took sun-synchronous measurements at 10 am - 10 pm equator crossing LST, in principle allowing to extend the validation of the diurnal climatology by means of observed day-night differences to different LSTs.

** This is an excellent idea, unfortunately we did not have time to acquire and familiarize ourselves with the MIPAS data. However we will continue to work to evaluate the diurnal model using additional data sources, and hope to collaborate on efforts with other instrument teams.

p11 l1-4: A possible reason for the divergence between GDOC and SAGE-III above 2hPa could also be related to the limitations of the monthly-resolved diurnal climatology: sunset (SS) and sunrise (SR) times are spread over a certain LST range in the monthly climatology, resulting in an artificial smearing of the diurnal gradient at SS and SR and hence in reduced SR/SS ratios.

** Yes, we now note this in the text: "At these levels, the SAGE III/ISS retrieval does not account for the sharp diurnal gradient in the ozone along the line of sight of the instrument. However, GDOC representations near the terminator may also be biased due to smearing of the diurnal ozone gradient in the monthly average as the terminator time shifts within the month."

p15 l4: the webpage is not accessible.

** The corrected web location has been added to the text.
* * *
[Figure]

[Figure]

**Fig. 1.**

---

## Author Comment (AC2) · 20 Feb 2020

Review of: Model-based Climatology of Diurnal Variability in Stratospheric Ozone as a Data Analysis Tool Stacey M. Frith1, Pawan K. Bhartia2, Luke D. Oman2, Natalya A. Kramarova2, Richard D. McPeters2, Gordon J. Labow The study is very detailed, and the results are convincing and new. For the first time, the authors demonstrate a feasible way how the effects of the diurnal ozone cycle in satellite and ground observations can be considered and partly removed. Thus, the article is of high interest for the readers of AMT. Future application of a related analysis to other diurnal cycles in other atmospheric parameters might be possible.

\*\* We thank the reviewer for their comments and address each point individually below,

as indicated by the bold text. We note that during the review process a model error was identified and a new simulation was run. We reanalyzed the new output, but found for ozone the differences were very small, and did not warrant producing a new climatology at this time. We will periodically update the climatology and include all model updates at that time.

I only found minor corrections which are listed below, and I have one question: I would be interested in the dependence of the diurnal cycle on longitude. Did you investigate if topography, convection or land-sea contrast have an influence on the diurnal cycle in the simulation data? Maybe you can add 1-2 sentences about this topic to your article.

\*\* Based on comments of another reviewer, we made a good faith effort to establish reasonable error estimates for GDOC, and in doing so did some analysis of the variability going into the averages, including variability in longitude. We found that the variability is quite large and complicated, and unpacking the sources of the variations will take some time. We cannot comment on this yet, but work is ongoing analyzing the model run. We note in the revised version of the manuscript that the uncertainty is largest in the high latitude winter, when the variability is greatest, which we associate with higher dynamical variability.

p.1, line 15 what is the meaning of GEOS-GMI?

\*\* The acronym has been expanded (and explained) in the abstract

p.2, line 4 Rowland instead of Roland

\*\* corrected, thank you

p.2, line 27 plural? Satellite data provide....

\*\* corrected

p.4, line 15 0.01 hPa instead of .01 hPa

\*\* corrected

p.4, line 20 please inform how the midnight value is defined, e.g., 23:00-1:00

\*\* The time resolution of GDOC (and the model output used to construct GDOC) is 30 minutes, thus the midnight time bin is 23:45-00:15. We have added this to the text.

p.8, line10 why did you change to the daily mean as reference?

\*\* In general, the reference point can be defined as any time in the cycle (or the daily mean), as is appropriate for the analysis. In this case the measurements are noisy, so normalizing to the daily mean demonstrates the similar structure in each data source but does not rely on the agreement at any single time. This was not made clear in the text and has been expanded upon.

\*\* In section 2.1 we added: "We note that GDOC can be re-normalized to any reference time as is most appropriate for a given analysis."

\*\* In Section 3.2 we added: Here we normalize to the daily mean rather than to values at a specific time in order to highlight the overall structure of the variability rather than differences at a single time.

p.8, line 19...measured by the satellite instruments.

\*\* corrected

p.11, line 3 line of sight?

\*\* Yes, corrected

p. 14, line 2...because no observational data source...?

\*\* corrected

p. 14, line 14 The sentence is not so clear. Perhaps "transits" instead of "transition"?

\*\* Thank you, we changed the wording to "shifts"

---

## Author Comment (AC3) · 20 Feb 2020

This manuscript describes the (GEOS-GMI) global model climatology (12 monthly sets) regarding ozone diurnal change as a function of local time for various latitude bins and pressure values. The chosen time step (resolution) is a half-hour. Model values are compared to various data sets, mostly from satellite-based ozone measurements with different spatio-temporal samplings. Most of the comparisons seem to validate the model results, even if there are a few discrepancies that are not completely explained. This model climatology is publicly accessible (or will be), and this offers a useful tool for other investigators, to try to improve certain upper stratospheric and mesospheric ozone comparisons.

[Figure]

** We thank the reviewer for their comments and address each point individually below, as indicated by the bold text. We note that during the review process a model error was identified and a new simulation was run. We reanalyzed the new output, but found for ozone the differences were very small, and did not warrant producing a new climatology at this time. We will periodically update the climatology and include all model updates at that time.

General Comments The paper is generally well-written, clear enough, and fairly thorough in the set of comparisons that are provided for validation. It does not purport to solve in detail every intercomparison's discrepancies. However, the lack of error bar discussion does raise some concerns, regarding the applicability for users; while the comparisons do indicate that the model provides a good representation of the true diurnal changes for ozone, the small differences that come up in terms of inter-instrument trend comparisons, for example, might still be "explained away" by uncertainties in model-based corrections, even after diurnal adjustments. Other uncertainties involve actual line-of-sight gradient issues, not just for the model results, but also for satellite-based retrievals, in particular, for solar occultation results (for which some attempts have been made to adjust for such gradients, but not as a general rule). These issues are the more difficult aspects, but this does not preclude, in my view, publication of this sort of manuscript. I ask for minor clarifications and some attempts (at least) at a better discussion regarding uncertainties, see my specific comments below. I also provide editorial-type comments, mostly as suggestions or corrections.

** We thank the review for their comments, and have added a good faith effort at reasonable error bars, as described further below. We also add some additional summary comments that try to clarify the conditions most applicable to GDOC: "We recommend using GDOC primarily for monthly zonal mean analyses in the pressure range from 30 to 0.3 hPa, and expect the climatology to capture diurnal variations to well within 5% in most cases. For finer resolution studies, GDOC can be used in a first order effort to estimate the impact of the diurnal cycle, to be followed by analyses that are more

refined. Users requiring more highly resolved information may contact the authors for access to the original model output."

Specific Comments One somewhat confusing detail has to do with the normalization time. For example, pg. 4, line 20, and pg. 6, line 4 refer to midnight as a normalization time. The Fig. 1 caption agrees with this description. However, the caption for Fig. 2 refers to 1:30 am as the normalization value, and so does Fig. S9. It would be good to clarify why there are these different normalization times, or if they should be the same. It probably does not matter too much, if different Figures are normalized slightly differently, but I found this confusing, so if something is written in error there, please correct.

** We have added a sentence that says GDOC can be normalized to any time (or to the daily mean time) as needed for a specific analysis. That being said, the normalization to 1:30am in the figures was a carry-over from some Aura MLS analysis, and is unnecessarily confusing as the reviewer points out. We have updated the plots to be normalized at midnight for consistency. In Figure 3 for example we deliberately chose to normalize to the daily mean, which, for the satellite data, was less noisy than the values at any particular time, and thus allowed for a better demonstration of the similar overall structures of the cycles, despite the noise.

2) Error bars are not always described (e.g., for Fig. 3), or justified (e.g., why not use standard error in the mean rather than standard deviation for Fig. 5 and Fig. 7, and similar Figures in the Supplement?). When using a very large data set (e.g., 2004-2018 Aura MLS data in Fig. 5), the random source of error will basically disappear. As an aside, geophysical variability probably accounts for some of the year-to-year differences; differences in day/night temperature or $H_2O$ ratios, for example, could have some impact on $O_3$ abundances and $O_3$ diurnal change. In the mid- to upper stratosphere, $N_2O$ day/night variability from year-to-year (or month-to-month) could impact ozone and its day/night ratios. Some comments about why the authors chose to use standard deviations rather than errors in the mean would be welcome (is it to

try to encompass such geophysical variability, which would be ignored in a standard error minimum type of error representation?). Maybe the standard deviation is indeed a more acceptable way to try to encompass sources of error, but I would welcome a brief comment regarding this point somewhere.

** We have added error bars and descriptions to all the relevant plots. In general, standard error of the mean is the statistic shown. In the Aura MLS profile day/night comparisons we wanted to highlight the year to year variability, so the standard variability is shown for the year to year variability only. This is now stated in the text. For GDOC itself we have added an uncertainty estimated based on the standard error of the mean. With so many measurements going in, the standard error of the mean was unreasonably small. We therefore computed correlations lagged in longitude to get an idea of the number of independent spatial measurements going into each bin, and based the standard error of the mean off that number (360 longitude points was reduced to 12). The resulting standard error of the mean varies up to 2%. We include the figure below as well as error bars on the GDOC profiles.

3) In some places, there is a mention of vertical "integration" of MLS data to match the vertical resolution of SBUV. This sort of smoothing is best done via the use of MLS Averaging Kernels (and MLS a priori data), although this can be somewhat tedious. The details are not mentioned here, but probably some indication of the "smoothing" or averaging process should be provided. Is there no smoothing in the Figure 5 results? Maybe errors in this, or omission of this, could lead to differences or discrepancies in the results (?). [It would also make more sense to smooth the MLS data sets for day and night and then calculate the ratios, than to smooth the MLS ratios, not that this is what was done].

** We typically use the SBUV averaging kernels applied to MLS, so that the degraded MLS correctly approximates the lower SBUV resolution. In this analysis we did not apply the averaging kernels, but we tested the results and found it does not make a difference. The impact of the averaging kernels is much smaller than that of the diurnal

correction. There is no smoothing in Figure 5. The MLS are simply interpolated onto the Z*star pressure grid. The following has been added to the data section: "OMPS NP and SBUV report ozone as partial column densities (in DU) in pressure layers. Number density and mixing ratio profiles are integrated to give cumulative column densities with pressure, which can be interpolated to re-partition the partial columns to match the SBUV/OMPS vertical sampling."

4) For Figure 6 in particular, the model could be used, in theory at least, to calculate line-of-sight differences in ozone signal for a solar occultation measurement, using small time steps for such a "ray-tracing" calculation, including height-dependence. Comparisons to a case assuming homogeneous line-of-sight ozone abundances, which is often assumed in retrievals, could be made. In theory, the sign of the differences in this case (model versus observations) could thus be ascertained. The authors could at least expand on this by stating that these comparisons are difficult because of not only the model calculation aspects but also the satellite retrieval aspects (they do mention the model, it seems, but not the satellite retrievals explicitly). It is alright to state that such detailed analyses are needed to better ascertain whether the model and data really disagree, even if the more detailed work is not pursued in this manuscript. Also, I wonder if one would not need finer sampling of the model in local time to match the fast changes at sunrise or sunset...(I am not asking to necessarily carry this research out in detail here).

** Yes, one would likely need a finer resolution diurnal information to untangle the diurnal impact on the occultation retrieval. This is actually the point we were trying to make (about the retrieval) but it was not clear in the original text. We have re-worded as follows: "At these levels, the SAGE III/ISS retrieval does not account for the sharp diurnal gradient in the ozone along the line of sight of the instrument. However, GDOC representations near the terminator may also be biased due to smearing of the diurnal ozone gradient in the monthly average as the terminator time shifts within the month. Also, as noted above, there is some variation between GDOC, WACCM and obser-

vations in the SR/SS pattern in the tropics. Nevertheless, these differences suggest potential discrepancies between SAGE III/ISS sunrise and sunset measurements that are currently being explored (R. Damadeo, personal communication, 2019). The purpose of this work is not to evaluate SAGE III/ISS observations but to demonstrate how GDOC can be used in such evaluations, even as a first-order indicator to a potential issue that requires more analysis."

5) Error bars: I would note that there are no error bars in Figure 6, so either they are too small, or just not calculated (as a standard deviation of the ratios, as done in other Figures), probably the latter. Including such error bars would make sense, however. Also, the error bars in Fig. 8 seem to be indicated by dashed lines, a different format, but please explain these ranges in the caption. Also, in Figure 9, maybe a standard error in the mean values as a function of time here would be more appropriate, but no error bars are shown; some comments regarding this (or actual error bars) would be appreciated as well. I expect that the volume of data used for these comparisons (for each specific month) is large enough to ensure that random errors become negligible.

** We have added error bars to Figure 6. In the case of the SAGE data, the standard error of the mean for the sunrise and sunset averages is computed, then the root mean square of the two errors is used as the final uncertainty (because of sampling it is the ratio of the average not the average of the ratios). In the case of GDOC, the model errors for each sub-sampled SAGE profile are collected and the root mean square of all the sunrise and sunset error profiles are computed first, then the root mean square of the resulting sunrise and sunset error profiles is computed, as was done for SAGE.

In Figure 9 we plotted the standard error of the mean, but it is smaller than the thickness of the line. We will add a comment in the caption, as well as indicate the size of the standard deviations. Editorial-type Comments / Suggestions-

** We thank the reviewer for their corrections and suggested changes, which made the manuscript read much better. Unless otherwise noted, we made all changes as

suggested. Page 1 L14, add a comma after "this issue". L16, change "applied in" to "applied to". - Page 2 L3, decide if use ODSs or ODS (I would follow the WMO Report type of writing, so probably ODSs for plural, elsewhere also) L6, change "has been" to "have been". L24, "to analyze the ozone diurnal cycle at ..." L28, change "Atmospheric" to "Atmosphere". - Page 3 L3, "non-sun-synchronous" L6, change "source" to "sources" .L7, I suggest "Also, these missions do not provide full global coverage." L25, "as well as to that from..." - Page 4 L19, it seems that "semi-hourly" should replace "hourly" here, since you use 30 minute model time steps.

\*\* We used the wording "half-hourly"

- Page 5 L7. You mention OMPS NP and OMPS LP. You also later refer to OMPS profile data and mention NP (top of page 12). Please clarify which data set is being used, NP or LP (or both?), as this was not quite clear enough; maybe this mainly requires a change on page 12. If the datasets are used as mentioned (LP for one plot, NP for another), please clarify (briefly) why one should use LP versus NP or vice-versa(what are advantages/disadvantages of NP versus LP?).

\*\* We have added some explanatory text. OMPS NP and LP are separate instruments but on the same platform; LP measures high resolution vertical ozone profiles, while NP is a nadir view instrument that measures in broad layers. Both are useful in their own right, so in this work we are demonstrating comparisons with both. The nadir instrument (OMPS NP) is best for total ozone and continues the SBUV nadir ozone record dating back to 1970. The LP provides very high resolution data, but as a new instrument it doesn't have the stable record of OMPS NP and its predecessors. We have added: "While OMPS LP is a limb scatter instrument that measures at high vertical resolution, OMPS NP is a nadir backscatter measurement with a broad vertical resolution in the stratosphere. Higher resolution instrument measurements (SAGE III/ISS, MLS, OMPS LP) are often used to help evaluate the lower resolution nadir instruments. This is critical to ensure OMPS NP can continue the 40+ year record of trend quality ozone from the SBUV series of nadir instruments."

- Page 6 L8, add commas "...very little, if any, variation..." L12, add a comma after "Parrish et al. [2014]". L15/16. However, SMILES data also suggest that ozone is decreasing..." L22, add a comma after "Figure 4a]". L28, either say "variations greater than" or "variations of more than" - Page 7 L2, authors suggest that the L8, Delete "Supplemental" L9/10, matches the higher summertime amplitude model diurnal cycle reported by Studer... L11, panels of Fig. 1 show the diurnal cycle... L14, change "greater" to "more". L18, but with larger afternoon values at 3 hPa L22, delete "Supplemental" [also, it is a bit strange to refer to S1 after you referred toS2 earlier] - Page 8 L20, relative maxima. L22, relatively high ozone value.

** Replaced sentence with "Finally at 5 hPa the stratospheric pattern dominates, with measurements and climatology showing the highest daily values in the mid-afternoon."

- Page 9 L7, add a comma after "this comparison" L8, shows the ratios of daytime to nighttime averages L17, amplitude of those in the MLS data, with ratios generally ...

** Replaced sentence with "Overall GDOC closely matches the spatial pattern and amplitude of the ratios measured by MLS, with agreement generally to within 2%." L18, near 1 hPa, we note a local minimum in ... L19, local minimum - Page 10 L2, delete "Supplemental" L23 and L26, (maybe) change middle latitudes to midlatitudes

** middle latitudes changes to mid-latitudes - Page 11 L3, change "site" to "sight" L18, add a comma after OMPS NP. also, please state briefly how the conversion for MLS O3 profiles from pressure to altitude is made.

** We have added the sentences "Aura MLS profiles are converted from volume mixing ratio on pressure surfaces to number density on altitude surfaces using co-located MERRA-2 temperature and pressure data." and "In this case SAGE profiles are converted from altitude to pressure using MERRA-2 pressure and temperature data provided with the SAGE III/ISS data." to this section.

L20, change "show" to "shows" L24, influence of the diurnal cycle on such analyses

[Figure]

- Page 12 L11, please add a sentence or two describing the "known bias pattern" for nadir UV instruments... Not everyone is familiar with what this means, and readers should not have to try to dig this out from other references (top-level information at least); "bias pattern" versus what? (in general?).

** We have re-worded this section as follows: "The remaining pattern of differences is consistent with biases previously reported in the nadir UV backscatter series of instruments relative to satellite (SAGE II, UARS and Aura MLS) and ground-based (select microwave and lidar) data [i.e. Kramarova et al., 2013; Frith et al., 2017]. Namely, the nadir backscatter instruments tend to have a negative bias below 10 hPa and above 2.5 hPa, and a positive bias near 7 hPa."

- Page 13 L10, please specify which instrument's results show a larger (or smaller) amplitude, is it MLS or SBUV, since the differences do not provide the reader with this information. One would think that the finer resolution instrument might provide a larger amplitude, although the broader vertical extent of the SBUV views means that this is actually not obvious.

** In this case, MLS is a reference instrument and does not change between the upper and lower plots. The SBUV has been "adjusted" to match the MLS measurement time of 1:30pm. We have added a statement in the text clarifying that only SBUV changes. L13, delete "Supplemental" L24, change ozone levels to ozone values L25, expressed as ratios to the value at midnight - Page 14 L11, change "depicts" to "exhibits" L27, suggesting that the representation... - Page 21, change Froidevaux to Froidevaux et al.; also change Livesay to Livesey. - Figure 1: the caption says "30 hPa to 0.3 hPa" but the plots seem to go down to 50hPa. Please clarify.

** The caption was incorrect, and has been corrected - Many of the Figures spell "AURA" rather than "Aura", which is the correct spelling (it is not an acronym), as spelled correctly in most of the manuscript. It would be good to correct the Figures for this.

** The spelling has been corrected in the figures. Also, Fig. 4, Fig. 5, and others in the Supplement have Day/Night Ratio as plot titles, but show ASC/DSC (for Fig. 5) in the axis labels... In reality, day and night during polar summer or winter does not make sense, as it is always either day or night, so it is more correct to state ASC/DSC as what is being calculated, if I am not mistaken. If this is true, the Day/Night labels should more properly be written as ASC/DSC, and for consistency with axis labels... At most latitudes, of course, this is the same thing...

** Yes, this is correct, ASC/DSC is the correct wording. We have changed to ASC/DSC throughout the text. In Fig. 5 (and others like it) there are confusing y-axis tick marks on the right side; it would be best to delete the altitude tick marks there.

** This has been corrected in all relevant figures. Also, in Fig.9, the last sentence could be rewritten a bit as "adjusted to *a* common time of 1:30pm, to coincide with *the* Aura MLS measurement time."

** Corrected Figure S10: change "output" to "outputs" in the last sentence; it would have been nice to indicate what mostly contributes to the differences between the 2017 and 2018 runs (is it geophysical variability in the model, or were there also some sampling differences in how this was calculated ,if matching SAGE sampling patterns?).

** We re-worded the last sentence as follows: "The final climatology is the average of output from the SAGE III 2017 and 2018 model runs." We have added the figure below showing the direct differences between the model simulation in 2017 and 2018 as a function of season and latitude at 4 pressure levels. The difference plotted is the max-min difference in local solar time. However, it is beyond the scope of the manuscript to examine the year to year differences beyond verifying that the year to year variability is small enough that the two can be averaged. There is no sampling difference between the two runs, the model is full spatial sampling.

[Figure]

[Figure]

**Fig. 1.**

[Figure]

**Fig. 2.**